

# Observed and simulated turbulent kinetic energy (WRF 3.8.1) over large offshore wind farms

Simon K. Siedersleben[1], Andreas Platis[2], Julie K. Lundquist[3,4], Bughsin Djath[5], Astrid Lampert[6], Konrad Bärfuss[6], Beatriz Canadillas[7], Johannes Schulz-Stellenfleth[5], Jens Bange[2], Tom Neumann[7], and Stefan Emeis[1]

[1]Institute for Meteorology and Climate Research (IMK-IFU), Garmisch-Partenkirchen, Germany
[2]Environmental Physics, ZAG, University of Tübingen, Germany
[3]University of Colorado, Department of Atmospheric and Oceanic Sciences, Boulder, Colorado, USA
[4]National Renewable Energy Laboratory, Golden, Colorado, USA
[5]Institute for Coastal Research, Helmholtz Zentrum Geesthacht, Germany
[6]Institute of Flight Guidance, Technische Universität Braunschweig, Germany
[7]UL-DEWI Gmbh

**Correspondence:** Simon K. Siedersleben (simon.siedersleben@kit.edu)

**Abstract.** Because wind farms affect local weather and microclimates, parameterizations of their effects have been developed for numerical weather prediction models. While most wind farm parameterizations (WFP) include drag effects of wind farms, models differ on whether or not an additional turbulent kinetic energy (TKE) source should be included in these parameterizations to simulate the impact of wind farms on the boundary layer. Therefore, we use aircraft measurements above large

5 offshore wind farms in stable conditions to evaluate WFP choices. Of the three case studies we examine, we find the simulated ambient background flow to agree with observations of temperature stratification and winds. This agreement allowing us to explore the sensitivity of simulated wind farm effects with respect to modeling choices such as whether or not to include a TKE source, horizontal resolution, vertical resolution, and advection of TKE. For a stably stratified marine atmospheric boundary layer (MABL), a TKE source and a horizontal resolution in the order of 5 km or finer are necessary to represent the impact

10 of offshore wind farms on the MABL. Additionally, TKE advection results in excessively reduced TKE over the wind farms, which in turn causes an underestimation of the wind speed above the wind farm. Furthermore, using fine vertical resolution increases the agreement of the simulated wind speed with satellite observations of surface wind speed.

## 1 Introduction

15 Offshore wind energy in Europe gains importance every year. In 2017 the wind energy market experienced a new record in investments with 3,148 MW additional net installed offshore energy equal to 560 new offshore wind turbines at 17 wind



farms. Two thirds of these turbines were installed in the North Sea, equal to an increase of 2,105 MW in net installed capacity (WindEurope, 2017).

Compared to onshore wind farms, offshore wind farms are larger in size, hence, the efficiency of large offshore wind farms is mainly driven by the turbulent vertical momentum flux (e.g., Emeis, 2010, 2018). Wind turbines extract kinetic energy from

the mean flow and convert it partly into electrical energy. The resulting wind deficit downwind is balanced by the advection of momentum of the mean flow and the turbulent momentum fluxes. Within large wind farms, the kinetic energy deficit is mostly balanced by the vertical momentum flux as the inner turbines are surrounded by wind turbines extracting the kinetic energy from the mean horizontal flow. Therefore, the wind speed reduction caused by the wind turbines upwind can be only balanced by the vertical momentum flux. Given generally low mean vertical velocities, the turbulent vertical momentum flux is crucial

when determining the power density of large offshore wind farms.

Global and regional climate simulations investigating the impact of wind farms have a horizontal grid too coarse to resolve wind turbines explicitly. Therefore, these studies are all based on models using wind farm parameterizations (WFP). In the past, areas of wind farms were represented as areas with increased surface roughness (Ivanova and Nadyozhina, 2000; Keith et al., 2004). Recently, wind turbines are parameterized as an elevated momentum sink at the levels intersecting with the rotor

area (Fitch et al., 2012; Volker et al., 2015). Additionally, the WFP of Fitch et al. (2012) adds turbulent kinetic energy (TKE) at the rotor area whereas the WFP of Volker et al. (2015) suggests that the TKE should be allowed to develop due to the resolved shear. However, both WFPs delivers similar results when calculating the power density (Volker et al., 2017).

Whether or not TKE enhancements should be included when using wind farm parameterizations to estimate impacts of wind farms is still a matter of debate. Several studies based on simulations suggest (e.g., Eriksson et al., 2015; Vanderwende

et al., 2016) that the wind farm parameterization of Fitch et al. (2012) adds too much TKE into the model causing exaggerated mixing while Vanderwende et al. (2016) also point out that removing TKE completely results in poor agreement with large-eddy simulations of wind farms. However, an accurate representation of observed TKE and the associated change in the vertical fluxes over a wind farm is difficult to evaluate. Therefore, implications of mesoscale wind farm parameterization on TKE were so far not evaluated.

In this study, we present aircraft observations taken approximately 60 m above large offshore wind farms at the North Sea in the framework of the project Wind Park Far Field (WIPAFF) (Emeis et al., 2016), measuring the wind speed and the TKE above two different wind farms. We use this data to evaluate the wind farm parameterization of Fitch et al. (2012) for three real case studies. More specifically we want to know:

– Do wind farm parameterizations for mesoscale models need a TKE source to resolve the enhanced TKE above wind
farms?

– How sensitive is the impact of the wind farm parameterization on the TKE to the horizontal and vertical grid of the driving model?

– How sensitive is the impact of the wind farm parameterization on the advection of TKE?





Section 2 gives an overview about the aircraft data, the configuration of the Weather Research and Forecasting model (WRF) (version 3.8.1) and the synthetic aperture radar satellite data (SAR) for our case study of 14 October 2017. Additionally, we summarize the wind farm parameterization of Fitch et al. (2012). We present the synoptic and atmospheric conditions during the three case studies in section 3. The measurements of the research aircraft are compared to the simulations in section 4,

followed by a sensitivity analysis with respect to the TKE source of the used WFP, horizontal and vertical resolution, and TKE advection in section 5. Recommendations for mesoscale offshore wind farm simulations are finally discussed in section 6.

## 2    Data and methods

Unique in situ aircraft measurements, described in section 2.1, allow evaluation of the simulated marine atmospheric boundary layer (MABL) in vicinity of the wind farms as well as TKE and wind speed over the wind farms. As we have a SAR (synthetic

aperture radar) satellite image for one case study a brief description of the SAR data is given in section 2.2. In section 2.3 the numerical model is presented followed by a description of the wind farms (section 2.4).

### 2.1    Aircraft measurements

The research aircraft Dornier 128-6, operated by the TU Braunschweig, flew with a true air speed of 66 m s$^{-1}$ on leveled and straight flight legs to measure wind speed, humidity, temperature and pressure at a sampling frequency of 100 Hz. Conse-

quently, our measurements have a horizontal resolution of 0.66 m (Platis et al., 2018). Wind speed observations have an relative error of 1% and 10% following Platis et al. (2018) and Siedersleben et al. (2018b) when averaging over 3000 and 300 data points, respectively, resulting in an error of ± 3° for the wind direction shown in the vertical profiles in section 3 (Siedersleben et al., 2018b). Details about the measurement devices installed on the Dornier 128-6 can be found in Corsmeier et al. (2001).

Three sets of aircraft observations are discussed herein, labeled as case I, II, and III, summarized in Table 1. The aircraft

observations were conducted on 09 August (case I), 14 October (case II) and 15 October 2017 (case III) at two different wind farm clusters (Fig. 1, and section 2.4). The observations above the wind farms on 09 August 2017 and 14 October 2017 started at ≈ 14:15 UTC and lasted 35 minutes and 52 minutes. The measurements on 15 October 2017 took place from 8:28 UTC to 09:21 UTC and 09:52 UTC to 10:17 UTC. The different observational periods are summarized in Table 1.

All aircraft measurements have the same pattern. Before we started the measurements over the wind farms, the aircraft

profiled the MABL in vicinity of the wind farms of interest, followed by several flights over the wind farms oriented perpendicular to the large scale synoptic forcing. During all observations the aircraft overflew the wind farm at least four times. Case study III included two additional measurements over the wind farms of interest conducted 40 min after the first four flight legs (Table 1).

The measurements were executed at two different wind farms (Fig. 1 and Fig. 2) with two different rotor types (more details

in section 2.4). Therefore, different flight heights were necessary - the aircraft flew at 200 m for case study I and 250 m for case study II and III over the wind farms (Fig. 3) Meerwind Süd Ost (MSO) and OWP Nordsee Ost (ONO), Godewind 1,2, (GW), respectively.



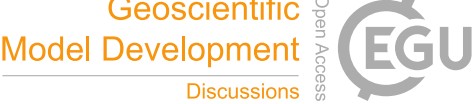

**Table 1.** Date, time, number of flight legs above the wind farms, location and flight height during the airborne observations

| Case study | Date | Time (UTC) | number of flight legs | wind farms | flight height (m AMSL) |
|---|---|---|---|---|---|
| I | 09 August 2017 | 14:14 - 14:51 | 4 | Meerwind Süd Ost, OWP Nordsee Ost | 200 |
| II | 14 October 2017 | 14:19 - 15:11 | 4 | Godewind 1, 2 | 250 |
| III | 15 October 2017 | 8:28 - 9:21 | 4 | Godewind 1, 2 | 250 |
| | 15 October 2017 | 9:52 - 10:17 | 2 | Godewind 1, 2 | 250 |

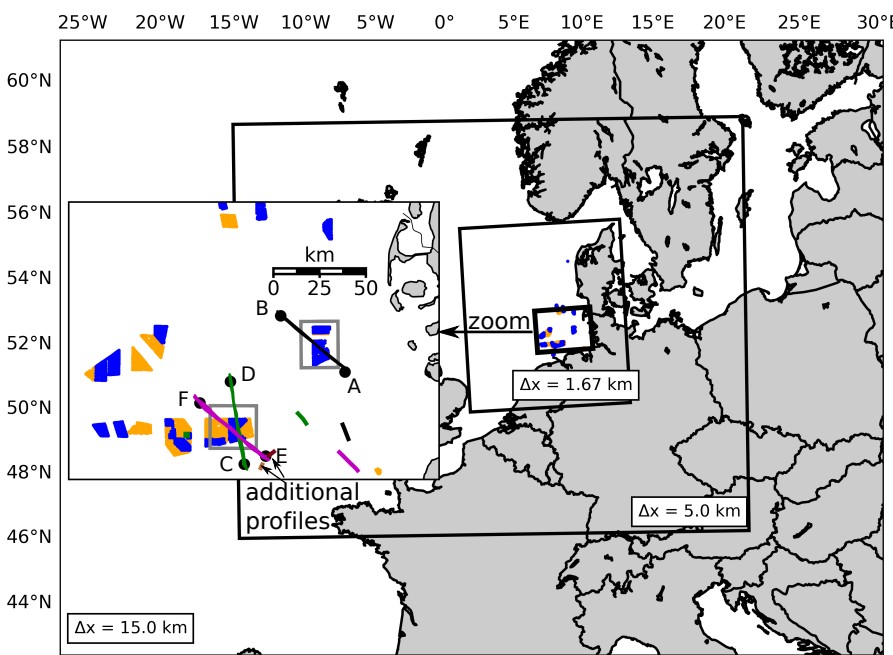

**Figure 1.** Locations of WRF domains and wind farms at the North Sea. A close-up on the German Bight shows the wind farms of interest framed with gray rectangles and the flight tracks of the three measurements in black, green and magenta, corresponding to the measurements executed on 09 August 2017, 14 and 15 October 2017, respectively. All measurements over the wind farms have a start and end point indicated with a capital letter for better orientation in Fig. 8, 10 and 12. Blue wind farms are in use, orange wind farms are approved or under construction according to plans in 2017. The thick lines indicate the locations of the climb flights, whereby the coloring corresponds to the coloring of the flight tracks over the wind farms, except the light red and red thick line showing the two additional profiles (see annotations) before and after the two additional flight legs. A detailed look (close-up on gray rectangles) on the wind turbine distribution of the wind farms of interest is provided in Fig. 2. The wind turbine location data was provided by the German Federal Maritime and Hydrographic Agency (BSH) and Bundesnetzagentur (2017).



## 2.2 Synthetic Aperture Radar Data (SAR)

Active remote sensing sensors, such as Synthetic Aperture Radar (SAR) have demonstrated the ability to provide 2D mapping of spatial variation of offshore windfarm wakes (e.g., Christiansen and Hasager, 2005; Li et al., 2014; Hasager et al., 2005, 2015; Djath et al., 2018; Ahsbahs et al., 2018) due to the large coverage and spatial resolution of few meters. Indeed, based on

Bragg scattering principle, SAR captures the small scale sea surface roughness, which is strongly related to wind conditions and returns the Normalized Radar Cross Section (NRCS). The combination of the C-band SAR satellite Sentinel-1A (launched in April 2014) and its twin Sentinel-1B (since April 2016) provides continuous measurements of the sea surface roughness of the German Bight with a repeat cycle of 6 days, but the same region can be sampled after one or two days with different incidence angle. Due to its sun-synchronous orbit, Sentinel 1 passes in the German Bight occur around 5 am or 5 pm. Figure 4

shows the 10-m wind speed derived from Sentinel1-A data acquired on 14 October 2017 at 17:17 UTC. The SAR scene is first calibrated to the NRCS using SNAP (Zuhlke et al., 2015) software supplied by European Space Agency. Then, the 10-m wind speed is derived using the geophysical model function CMOD5N (Hersbach et al., 2007; Verhoef et al., 2008) and hourly wind direction from German Weather Service (DWD) forecast model. The detailed methodology for the data processing and wind retrieval is described in Djath et al. (2018).

## 2.3 Numerical Setup

All simulations were performed with the Weather Research and Forecasting Model WRF (version 3.8.1) (Skamarock et al., 2008). We used three domains with 16 km in the outermost domain followed by two domains with 5 km and 1.67 km, respectively (Fig. 1). The boundary and initial conditions were provided by ERA5 data (Copernicus Climate Change Service , C3S), with horizontal resolution of 0.25 $^\circ$ and 138 vertical levels.

All simulations were initialized the night before the observations at 00:00 UTC, resulting in a spin-up time of more than 12 hours for case study I and II as suggested by Hahmann et al. (2015). Case study III took place earlier resulting in a spin up time of only eight hours.

We use two different sets of vertical levels. The default configurations corresponds to that of Siedersleben et al. (2018b, a) with a vertical spacing of 35 m in the lowest 200 m and increasing to 100 m at 1000 m above mean sea level (AMSL)

corresponding to one vertical level below the rotor area and three within the rotor area for the wind turbine type installed at the wind farms MSO and ONO (case study I, Fig. 3). Four vertical levels are located within the rotor area for the wind farm GW (Fig. 3) due to the larger rotor area. Lee and Lundquist (2017) obtained best results with 80 vertical levels - equal to a vertical spacing of 12 m below 400 m AMSL. Therefore, we tested the sensitivity of our results using the vertical levels of Lee and Lundquist (2017) - equal to three full levels below and ten full levels within the rotor area for the wind farms MSO, ONO (case

study I) and two full levels and 13 full levels within the rotor area for the wind farms GW 1, 2 (case study I and II, Fig. 3).

We use the WFP of Fitch et al. (2012) to represent the wind farms in WRF. The parameterization extracts kinetic energy from the mean flow and adds TKE at the vertical levels intersecting with the rotor area, depending on the thrust- and power coefficients of the wind turbines. The thrust coefficient is the fraction of energy extracted from the mean flow, the power

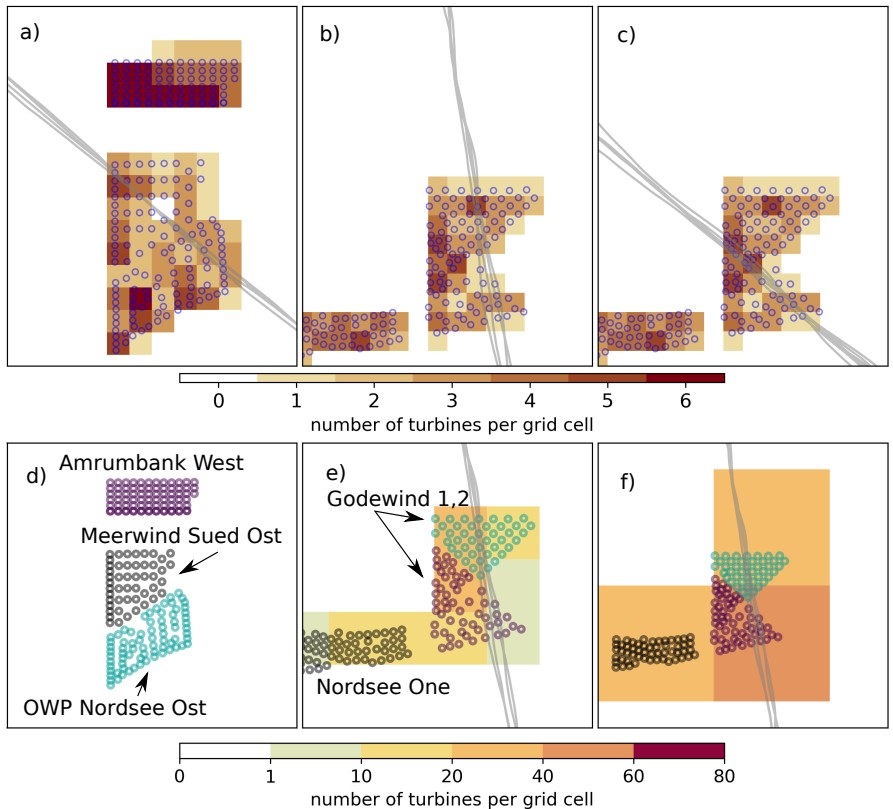

**Figure 2.** The number of wind turbines within one grid cell in colored contours for the wind farms (a) Meerwind Süd Ost (MSO) and OWP Nordsee Ost (ONO) and (b-c) Godewind Wind 1,2 (GW) for the control simulations (CNTRa, CNTRb, CNTRc). The size of the contour areas correspond to the size of the horizontal model grid. The circles denote the exact locations of the single wind turbines whereby the wind turbines are colored according to the wind farm they belong to in (d-f), additionally (e-f) show the horizontal grid with 5 km and 16 km resolution for the sensitivity studies: DX5, DX16, DX5noTKEsource and DX16noTKEsource. The wind turbines are not colored in (a-c) for better visibility of the wind turbine density. The gray lines denote the flight track of the research aircraft.

coefficient is the fraction of energy converted into electrical energy. These coefficients depend on the wind speed and wind turbine type because different wind turbine types are designed for different wind regimes.

The WFP of Fitch et al. (2012) neglects electrical and mechanical losses and assumes that all non-productive drag is converted into TKE. Consequently, the difference between the thrust- and power coefficient describes the fraction of energy that is converted into TKE. More specifically, the amount of TKE added to the model is:

$$\frac{\partial TKE_{ijk}}{\partial t} = \frac{\frac{1}{2} N_t^{ij} C_{TKE}(V_H) V_{ij}^3 A_{ikj}}{z_{k+1} - z_k} \tag{1}$$

$$C_{TKE} = C_T - C_P \tag{2}$$

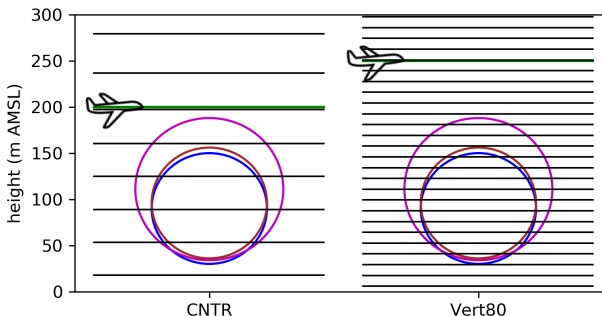

**Figure 3.** Distribution of the vertical levels with height and the levels intersecting with the rotor areas of the two wind turbine types used in the wind farms as listed in Table 3 for the CNTR and the Vert80 simulation. The rotor areas of the wind turbines SIEMENS-SWT-6.0-154, SIEMENS SWT 3.6-120 and SENVION 6.2 is shown in magenta, blue and red, respectively. The green lines denote flight heights at 200 m and 250 m AMSL; necessary due to the two different wind turbine types.

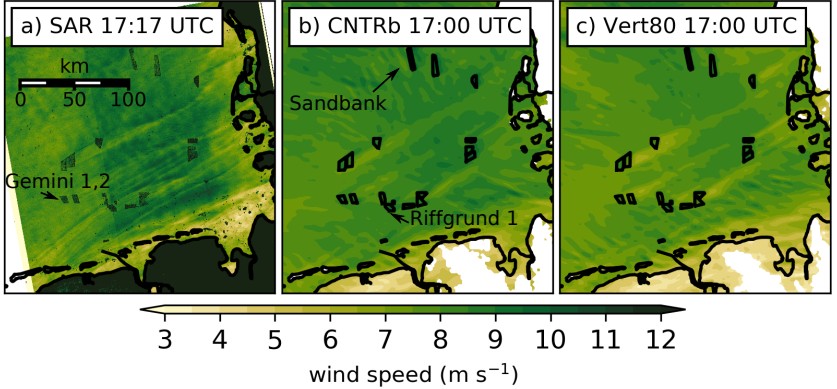

**Figure 4.** A comparison of SAR from Copernicus Sentinel 1A and WRF retrieved wind speed at 17:17 UTC and 17:00 UTC, respectively, on 14 October 2017 (case study II). The SAR data shows the wind speed at 10 m, whereas the model output is taken from the model level closest to 10 m. Therefore, we show the wind speed in 17 m and in 15 m for the CNTR and Vert80 simulation, respectively.

whereby $C_T$ and $C_P$ are the thrust- and power coefficients and $C_{TKE}$ is the fraction of energy converted into TKE. Equation (1) is formulated for a Cartesian coordinate system with the indexes i, j, k corresponding to the directions x, y, z, that in turn is equal to the geographic directions West-East, South-North, and the vertical axis with k=0 the level closest to the ground. The variable $N^{ij}$ describes the number of wind turbines within a grid cell $i,j$; $V_{ij}$ is the horizontal wind speed at hub height at grid cell $ij$ (Redfern et al. (2019) showed that during strong shear events rotor-equivalent wind speed result in a different change in TKE). $A_{ikj}$ is the rotor area between the two vertical levels $k$ and $k+1$, at a height $z_k$ and $z_{k+1}$, and $V_H$ the horizontal wind speed at hub height. Consequently, the increase in TKE with time is highest for high wind speeds at hub height, high number of wind turbines in one grid cell and a large difference between the power and the thrust coefficient.





A summary of our sensitivity tests appears in Table 2. The sensitivity of our results with respect to the horizontal grid size was tested with simulations of 5 km and 16 km horizontal resolution, respectively. Consequently, the number of turbines within one grid cell changes (Fig. 2). In Siedersleben et al. (2018a, b) we obtained best results using a horizontal grid size of 1.67 km.

The two sets of vertical levels demand two different time step configurations. For the default configuration we use a time step

of 60 s in the outermost domain, 20 s in the second domain and 5 s in the third domain. For the simulations using refined vertical levels, smaller time steps are necessary due to the higher resolution. Therefore, we use 10 s, 3.33 s and 0.67 s corresponding to the three domains. We named the simulations using 80 vertical levels Vert80.

Recently, some published studies (e.g., Abkar and Porté-Agel, 2015; Eriksson et al., 2015; Vanderwende et al., 2016; Pan and Archer, 2018) suggested that the mixing induced by the WFP of Fitch et al. (2012) is too high due to the added TKE

into the model (see Eq. 1). Therefore, we tested the sensitivity of our simulations by switching the TKE source off. Three simulations were performed using a horizontal grid spacing of 1.67 km, 5 km and 16 km with a disabled TKE source (noTKE-source, DX5noTKEsource, DX16noTKEsource). As we expect a simulation with more vertical levels to resolve more vertical shear, we performed additionally a simulation using 80 vertical levels with a grid size of 1.67 km and no TKE source (Vert80noTKEsource).

Since WRF version 3.5.0, TKE advection can be activated in the boundary scheme of Nakanishi and Niino (2006) (see Eq. 1.4 in Skamarock et al. (2008)). In previously published studies (e.g., Mangara et al., 2019) this option was used. Therefore, we tested the sensitivity of our results with respect to this option. A summary of all sensitivity tests is shown in Table 2.

Public information on turbine thrust and power coefficients is not widely available, and so we also explored the sensitivity of our results to these parameters. We altered the estimated thrust coefficient similar to Siedersleben et al. (2018b) by ±10 %,

resulting in two simulations (ThrustMinus, ThrustPlus) that are expected to introduce more and less TKE into the model than the CNTRb simulation. The results are presented in section 5.4.

The following parameterizations were used: the WRF double-moment 6-class cloud microphysics scheme (WDMS; Lim and Hong (2010)), the Rapid Radiative Transfer Model for GCM (RRTMG) scheme for short- and longwave radiation (Iacono et al., 2008), the Noah land surface model (Chen and Dudhia, 2001) and the Mellor-Yamada-Nakanishi-Niino (MYNN) boundary

layer parameterization (Nakanishi and Niino, 2006) interacting with the WFP of Fitch et al. (2012). Only the first domain uses the cumulus parameterization of Kain (2004) as the two innermost domains have a convection-permitting resolution.

### 2.4 Wind farms at measurement sites and in the mesoscale model

We conducted the three measurements at two different sites. Case study I took place at the wind farms Meerwind Süd Ost (MSO) and OWP Nordsee Ost (ONO) (Fig. 2a, d), whereas the other two case studies focus on Godewind 1 and 2 (GW)

(Fig. 2b,c).

The wind farms MSO and ONO are equipped with SIEMENS SWT-3.6-120 and SENVION 6.2 turbines having a hub height of 90 m and ≈ 95-97 m, respectively; a rotor diameter of 120 m, resulting in a rotor top of 150 m and 157.4 m at most. GW are equipped with Siemens SIEMENS SWT-6.0-154 wind turbines having a hub height of 110 m and diameter of 154 m, resulting in a rotor top of 187 m. Further details are provided in Table 3.





**Table 2.** Overview of performed numerical simulations and parameter choices for the sensitivity experiments.

| simulation | horz. grid size (km) | vertical levels | TKE source | TKE advection | thrust coefficient |
|---|---|---|---|---|---|
| CNTRa | 16, 5, 1.67 | 50 | on | off | default |
| CNTRb | 16, 5, 1.67 | 50 | on | off | default |
| CNTRc | 16, 5, 1.67 | 50 | on | off | default |
| DX5 | 16, 5 | 50 | on | off | default |
| DX16 | 16 | 50 | on | off | default |
| Vert80 | 16, 5, 1.67 | 80 | on | off | default |
| noTKEsource | 16, 5, 1.67 | 50 | off | off | default |
| Vert80noTKEsource | 16, 5, 1.67 | 80 | off | off | default |
| DX5noTKEsource | 16, 5 | 50 | off | off | default |
| DX16noTKEsource | 16 | 50 | off | off | default |
| ADV | 16, 5, 1.67 | 50 | off | on | default |
| ThrustPlus | 16, 5, 1.67 | 50 | on | off | $+10\,\%$ |
| ThrustMinus | 16, 5, 1.67 | 50 | on | off | $-10\,\%$ |

**Table 3.** Wind turbine types installed at the measurement sites according to the data of the Bundesnetzagentur (2017)

| wind farm | wind turbine type | hub height (m) | diameter (m) | rotor top (m) | number of wind turbines |
|---|---|---|---|---|---|
| Godewind 1, 2 | SIEMENS SWT-6.0-154 | 110 | 154 | 187 | 97 |
| Meerwind Süd Ost | SIEMENS SWT 3.6-120 | 90 | 120 | 150 | 74 |
| OWP Nordsee Ost | SENVION 6.2 | 95.4-97.04 | 126 | 156 | 48 |

In this study, we use as in Siedersleben et al. (2018b), the thrust- and power coefficients of the wind turbine SWT 3.6-120-onshore, as these are freely available online[1]. We use these power and thrust coefficients for all wind turbines implemented in the model. However, the hub height and the rotor diameter was adapted for each wind turbine type in the simulations. The locations of the installed wind turbines (i.e. all blue wind turbines in Fig. 1) were taken from Bundesnetzagentur (2017).

5 **3 Observations**

Here we use ERA5 data to provide overviews of the synoptic situations before and during each case study (Copernicus Climate Change Service , C3S). Additionally, the vertical structure of the atmosphere is discussed by the use of the climb flight data. Finally, the results of the aircraft measurements over the wind farms for the three case studies are described in section 3.2 and 3.3.

---

[1]http://www.wind-turbine-models.com/turbines/646-siemens-swt-3.6-120-onshore, accessed on 16.01.2018)





### 3.1 Synoptics and mesoscale overview

Case study I was stably stratified with wind from the south west. On 09 August 2017 at 15:00 UTC, a trough approached the German Bight from the North (Fig. 5a) associated with southwesterly winds of 10-12 m s$^{-1}$ at hub height near MSO (Fig. 6a, g). Warm air advection was associated with a stably stratified atmosphere according to the climb flight (Fig. 6g)

upwind of the wind farm cluster (Fig. 1, black thick line). Despite the stably stratified atmosphere, the sea surface temperature (SST) was warmer than the air temperature close to the surface. At the FINO1 tower a SST of 292 K was measured, ≈2 K warmer than the air temperature. As expected for summer time, the SST was highest closest to the coast (Fig. 5a).

Case study II was also stably stratified, with stronger winds from the west. On 14 October 2017 at 15:00 UTC, a deep trough located over the Atlantic caused a zonal jet over the North Sea (Fig. 5b) associated with wind speeds of up to 15 m s$^{-1}$ at hub

height (Fig. 6e) at the location of the climb flight (Fig. 1, green thick line). Due to the stably stratified atmosphere, the wind profile was characterized by strong vertical shear between 30 m AMSL and 190 m AMSL (Fig. 6e), corresponding to the rotor area limits of the wind farm. According to the SAR data, the stably stratified atmosphere was associated with wakes longer than 50 km (Fig. 4a). Long wakes are visible downwind of the wind farms located near the German and Netherlands coasts. Further to the north, around the wind farm Sandbank (see annotation in Fig. 4a) only subtle wakes are visible indicating less

favorable conditions for wakes.

Case III also experienced a stably stratified flow with 10 m s$^{-1}$ wind speed and southerly wind direction (Fig. 6c,f). On 15 October 2017 (case III) the trough over the Atlantic moved further to the west causing a south-westerly warm air advection, that in turn resulted in a pronounced inversion with a temperature difference of 4 K between 30 m AMSL and 190 m AMSL according to the profile recorded by the aircraft (Fig. 6c, magenta thick line in Fig. 1). Associated with the top of the inversion

is a wind speed maximum at ≈ 190 m AMSL (Fig. 6f). From previous literature (e.g., Smedman et al., 1997; Dörenkämper et al., 2015; Svensson et al., 2016) we would expect a SST colder than the air temperature close to the sea surface. However, a SST of 288.5 K was measured at FINO1 in contrast to a potential air temperature of 285 K at 50 m AMSL according the airborne measurements, indicating that the SST was warmer than the air temperature. The two additional vertical profiles taken before and after the additional flyovers revealed a destabilization of the atmosphere (Fig. 6c).

### 3.2 Wind speed over and next to the wind farms

For case study I, wind speeds in the order of 13 m s$^{-1}$ were observed at 200 m AMSL near the wind farms (Fig. 7a, Fig. 8a). During the four flights over the wind farms the wind speed varied only by ± 0.5 m s$^{-1}$ indicating that the weather situation was stationary. However, the variability of the wind speed measurements increases over the northern edge of the wind farm ONO. At the downwind side of the wind farm, the wind speed decreased in each observation by more than 1 m s$^{-1}$, indicating

that the wake of the wind farm extended to a height of 200 m. As the aircraft approached the upwind side of the wind farm (i.e. at 54.46 °N latitude) the wind speed deficit decreased.

During case study II, the distinct wind farm wake was also accompanied by a speed-up around the farm, such as indicated by Nygaard and Hansen (2016). We observed a horizontal wind speed of ≈15 m s$^{-1}$ at 250 m AMSL south of GW and slightly



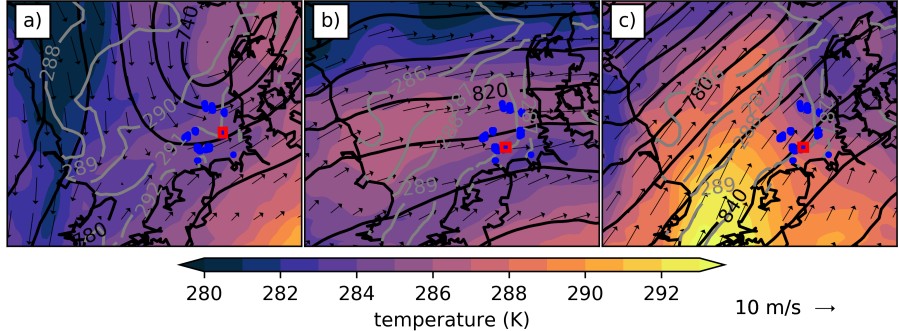

**Figure 5.** ERA5 reanalysis data: Temperature (colored contours (K)) and geopotential height (20-m increments) as black contour lines in 925 hPa for the three case studies as listed at Table 1, at 15:00 UTC 09 August 2017, at 15:00 UTC 14 October 2017 and at 09:00 UTC 15 October 2017 (Table 1). The gray solid contour lines show the SST.

lower winds speeds to the north (Fig. 8b, Fig. 9b). At the southern edge of the wind farm, orientated parallel to the large scale synoptic forcing, the wind speed dropped consistently in all four flight legs by up to 2 m s$^{-1}$, associated with a speed up further south (see annotation of Fig. 8b). We suggest that this acceleration emerges due to an enhanced flow around the wind farm due to the stably stratified atmosphere. Similar to case I, the wind speed showed low variability during the measurements that were
performed within a time interval of 50 minutes; highest variability occurred over the wind farms.

In comparison to the case studies I and II, in III the wind speed was barely influenced by the wind farms GW during the first flight legs. We suggest that this phenomenon is rooted in the strong inversion between 40 m and 180 m AMSL, decoupling the layer over the inversion from the surface layer. Consequently, the wind speed measurements showed only weak enhanced variability over the wind farms. However, two additional measurements were taken 40 minutes later. These two flyovers both
show an enhanced deceleration over the wind farms, especially the last flight leg (purple line, Fig. 8c). The mean (shown in Fig. 8c) was calculated using only the first four flight legs.

### 3.3   TKE over and next to the wind farms

In case I, the airborne measured TKE over the wind farm was a factor of ten higher than in the ambient flow. The TKE over the wind farms MSO and ONO was increased compared to the surrounding (Fig. 9a, Fig.10a), over the wind farms the research
aircraft measured a TKE of up to 2.0 m$^2$ s$^{-2}$, but 0.2 m$^2$ s$^{-2}$ within the undisturbed environment, meaning that the TKE over the wind farms is almost ten times higher 50 m over the rotor top compared to the surrounding environment. This pattern was observed during all four flyovers (Fig. 10a). The mean of all measurements clearly indicates that the highest TKE was observed in the wake region of the wind farm MSO where the shear was greatest (shown in Fig. 8a).

In case II, TKE over the wind farms was a factor of 20 times higher than in the ambient flow. A TKE of up to 2.5 m$^2$ s$^{-2}$ was
observed at 250 m AMSL over the wind farms GW and 0.1 m$^2$ s$^{-2}$ within the background flow (Fig. 9b). The TKE maximum,



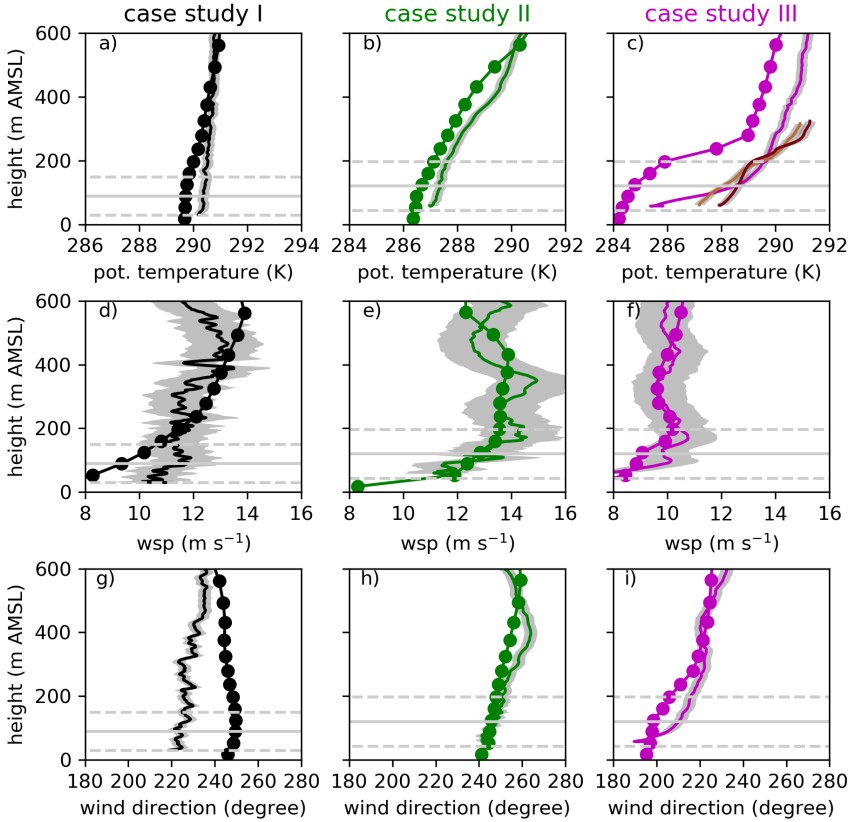

**Figure 6.** Vertical profiles of potential temperature (a-c), wind speed (d-f) and wind direction (g-i) obtained by probing the atmosphere with the research aircraft (solid lines). The interpolated WRF data along the climb flight is shown with the line having the circles on top, whereby each circle represents a vertical level of the WRF control simulation (CNTRa, CNTRb, CNTRc). The gray shadings represent the error bars of the measurements. The dashed and solid gray lines denote the rotor area and the hub height of the wind turbines. As the measurements were conducted at two sites with two different wind turbine types, the height of the hub and rotor areas vary. In (c) two additional vertical profiles are shown (red and light red) that were taken before and after the additional flyovers in case study III, for further details see text. Each column corresponds to one case study, i.e. the column (a, d, g) corresponds to case study I, similar to the coloring of flight tracks in Fig. 1.

visible in all four flights (Fig. 10b), corresponds to the southern edge of the wind farms GW - the region with the highest horizontal wind shear (Fig. 8b). In contrast, no TKE maximum can be observed upwind at the northern edge of the farm GW.

   In case III, a strong inversion generated a stably stratified environment resulting in lowest TKE values observed within our three case studies in the background flow and over the wind farm (Fig. 9c and Fig. 10c). Nevertheless, the values of TKE
5 over the wind farms during all six flights were elevated compared to surroundings. The TKE maximum matched in location with the western edge of the wind farm where the horizontal wind shear was greatest (Fig. 8c) due to the south westerly background flow. During the last flight leg, the aircraft observed TKE in the order of 1.6 m$^2$ s$^{-2}$ , three times higher than in





the measurements conducted 40 minutes before (Fig. 10c). This specific flight leg showed also the strongest wind deceleration over the wind farm (Fig. 8c).

In every case, above the wind farm, the aircraft observed values of TKE between five and 20 times larger than the ambient values of TKE.

## 4 Control simulations

Herein, we present control simulations for each of the three case studies I, II and III. We start with a comparison of the vertical profiles of the aircraft measurements and the profiles obtained by the simulations. As we want to evaluate the TKE over the wind farms that is in turn highly dependent on wind shear, we compare the wind speed measurements with simulations before we evaluate the simulated TKE in section 4.3.

### 4.1 Evaluation of the background flow

For case study I, the simulated potential temperature profile and the observations show a weakly stratified atmosphere (Fig. 6a), whereas the model is more stably stratified between 90 m and 250 m AMSL, resulting in stronger vertical wind shear in the model (Fig. 6d). An explanation for this behavior could be the more south westerly winds in the model causing a stronger warm air advection (Fig. 6g). However, warm air advection should be associated with a veering wind. In contrast, the veering is more pronouced in the observations (Fig. 6g).

For case study II, the observed and simulated vertical structure of the atmosphere agree except for a cold bias in the potential temperature. The model predicts a potential temperature profile with a lapse rate similar to the observed one but with a cold bias of 0.5 K. The strong vertical wind shear within the lower rotor area is well represented, so is the wind direction. Consequently, the orientation of the wakes in the SAR satellite observations (Fig. 4a) match with the simulated wakes in Fig. 4b). Note, the SAR image taken on 14 October 2017 at 17:17 UTC should be only used to evaluate the orientation of the wakes. The lowest level of the control simulation is at 17 m AMSL. Consequently, interpolating the wind speed to a height of 10 m is difficult. Therefore, we show the simulated wind speed at 17 m in Fig. 4b) for simplicity.

For case study III, the simulations show a less pronounced inversion than the observations (Fig. 6c). This behavior of the model is similar to the case study presented in Siedersleben et al. (2018b), where an inversion similar to the one shown in Fig. 6c) developed and the WRF model had also problems to represent the inversion. In this case the inversion is even more pronounced, most likely associated with the proximity of the vertical profile to the coast (Fig. 1, thick magenta line), increasing the challenge for the model to capture the heterogeneity. However, this inversion weakened during the observation but the stratification of the atmosphere in the model did not change with time. Therefore, the simulated profiles before and after additional flights are not shown in Fig. 6c.

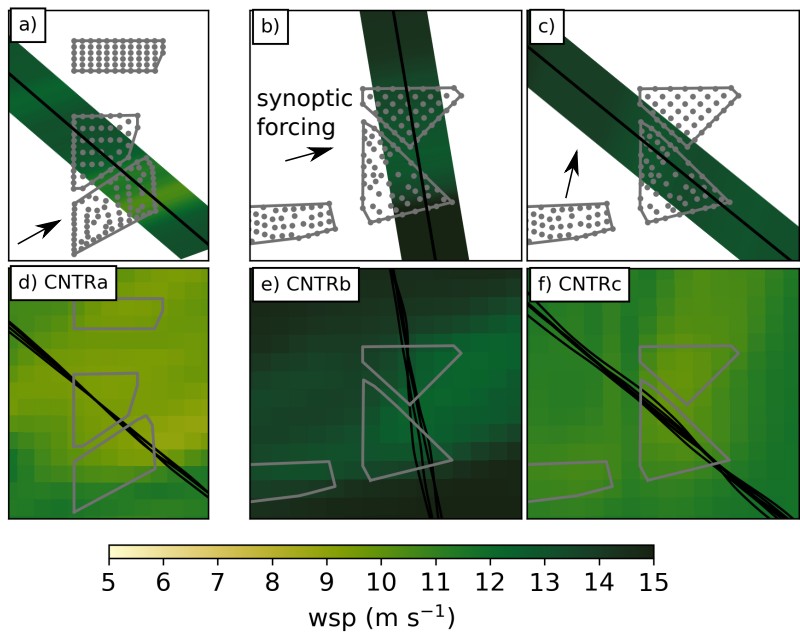

**Figure 7.** Plan view on observed (a-c) and simulated (d-f) horizontal wind speed at 14:30 UTC, 15:00 UTC and 09:00 UTC on 09 August, 14 and 15 October 2017, respectively, horizontal wind speed in colored contours at 200 m AMSL (a, d) and 250 m AMSL (b, c, e, f). Black lines denote the flight path over the wind farms. The observations show the mean of the observed wind speed, similar to Fig. 8a-c). The locations of wind farms and single wind turbines are shown by gray polygons and dots, respectively. Each column corresponds to one case study (i.e. column a,d corresponds to the measurements taken on 09 August 2017).

## 4.2 Impact of wind farm parameterization on wind speed above wind farm

The simulation for case study I generally underestimates the wind speed at 200 m AMSL over and next to the wind farms by up to 2 m s$^{-1}$ (Fig. 7a, Fig. 8a). The sharp decrease of 1 m s$^{-1}$ within the wake is captured by the model at 15:00 UTC but not at the beginning of the measurements at 14:30 UTC. A weak increase in wind speed similar to the observation is

5   represented over the wind farm (i.e. within the gray shaded area in Fig. 8a), associated to the shorter distance of the upwind edge of the wind farm. A possible explanation for the wind speed bias between model and observation could be a more unstably stratified atmosphere in the simulations. However, the model adequately represented the stratification of the atmosphere in the vicinity of the wind farms (Fig. 6a). Therefore, we suggest that the atmosphere was more stably stratified to the west during the observation as in the simulations.

10   The simulations for case study II represent the stationary background flow (i.e. no variance between 14:30 UTC and 15:00 UTC) and the impact of the wind farms GW well. The averaged wind speed matches with the simulated wind speed within ±0.2 m s$^{-1}$, except at the southern edge of the wind farm - there the horizontal wind speed gradient is more pronounced



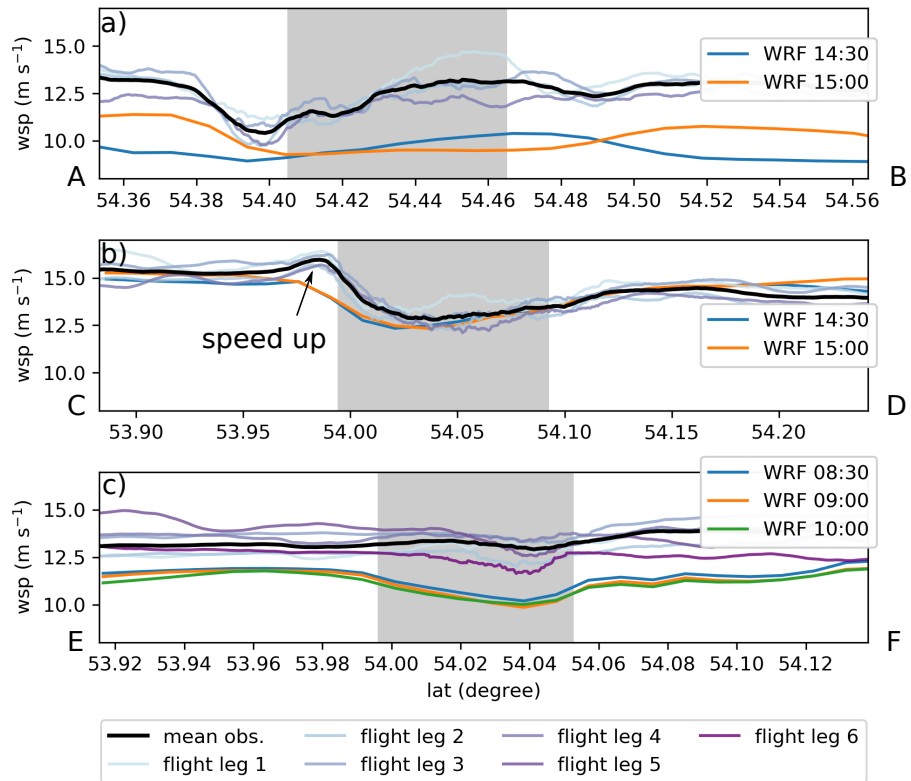

**Figure 8.** Observed (thin blue and purplish lines, the purple amount is increased, the later the flight leg was flown) and simulated wind speed interpolated onto the flight track in blue for the three case studies (a-c) as indicated in Table 1. The black thick line shows the mean of all wind speed measurements over the wind farms similar to the measurements shown in Fig. 7a-c). The gray shaded areas denote the location of the wind farm. The capital letters on the x-axis show the orientation of the axis as indicated in Fig. 1.

in the observations than in the simulations. However, this deviation is likely rooted in the rather coarse horizontal grid size of the model.

The model underestimates the wind speed compared to the measurements conducted during case study III. Over the wind farms the deviation between simulations and observations are largest for the first four flyovers, indicating a more pronounced impact of the wind farms on the atmosphere in the simulations than in the observations. However at 10:00 UTC, the observation showed an increased impact on the wind speed over the wind farms similar to simulations with a constant negative bias of $\approx 2.0$ m s$^{-1}$.

### 4.3 Impact of wind farm parameterization on TKE above wind farms

The increased TKE over the wind farms is captured by the simulations, but not the shape of the TKE profile over the wind farms. For example in case study I, the WFP simulates a TKE over the wind farms with two peaks (Fig. 9a, Fig. 10a), whereby

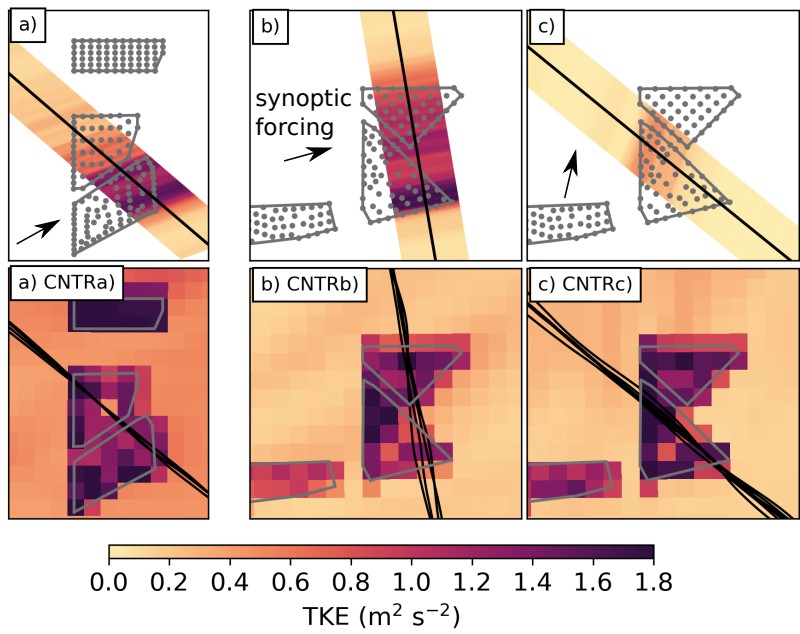

**Figure 9.** As Fig. 7, but for the TKE.

the first peak matches with the observed TKE maximum with $\approx 1.5$ m$^2$ s$^{-2}$. However, this peak in TKE corresponds in the observations to the southern edge of the wake that developed behind the farm, whereas in the simulations this peak corresponds to the southern edge of the wind farm (Fig. 2a). The second peak at 54.46°N in simulations with a TKE of 2.0 m$^2$ s$^{-2}$ corresponding to the upwind side of the wind farm was not observed.

5  A similar pattern can be observed for the simulations conducted for 14 October 2017. The TKE maximum at the southern edge of the wind farms is captured by the model (Fig. 9b, Fig. 10b). In contrast, the declining trend of TKE towards the northern edge of the wind farm is interrupted in the model. The TKE of the undisturbed flow next to the wind farm is very similar to the observed TKE, increasing the confidence in this simulation.

In contrast to the other case studies, in case study III, the TKE in the observations evolves over time. Initially, the simulated

10  TKE is more than twice as high than the averaged observed TKE over the wind farms, 1.0 m$^2$ s$^{-2}$ for the first four flyovers compared to 2.0 m$^2$ s$^{-2}$ (Fig. 9c, Fig. 10c). However, 40 minutes later, the measured TKE from the additional two flight legs show a TKE similar to the simulations. Especially, the last flight leg shows a TKE of 2.0 m$^2$ s$^{-2}$ at the western edge of the wind farm. This flight leg also has the most pronounced wind speed deficit over the wind farm agreeing best with simulated impact on the horizontal wind speed at 250 m AMSL.



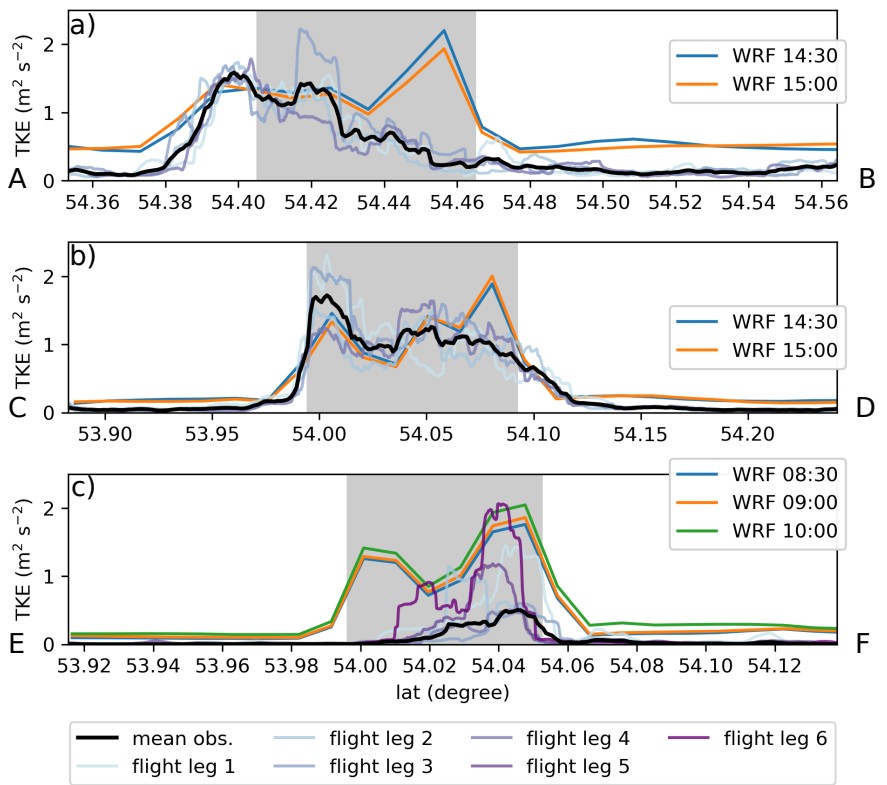

**Figure 10.** As Fig. 8, but for the TKE.

## 5 Sensitivity experiments

In case II, the model captures the background flow providing a sound basis for sensitivity studies. In contrast, the simulations for case study I and III have both a bias in the wind speed at 200 m or 250 m over and next to the wind farms associated with a deviation considering the intensity of an inversion for case III. For case I, we can only suggest that the negative bias in the horizontal wind speed is rooted in the stratification of the model due to the lack of measurements available at the North Sea.

Given the success with the simulation CNTRb, we explore the sensitivity of the WFP of Fitch et al. (2012) with respect to horizontal grid size, the TKE source, vertical resolution, TKE advection and thrust coefficient in sections 5.1, 5.2, 5.3 and 5.4.

### 5.1 Sensitivity to horizontal and vertical resolution with an active TKE source

We conducted two additional simulations with a horizontal grid size of 5 km and 16 km with the TKE source of the WFP of Fitch et al. (2012) active; these simulations are called DX5 and DX16. Additionally, a third simulation was performed with the same configuration as CNTRb but with 80 vertical levels (Vert80). A summary of all sensitivity tests is given in Table 2.



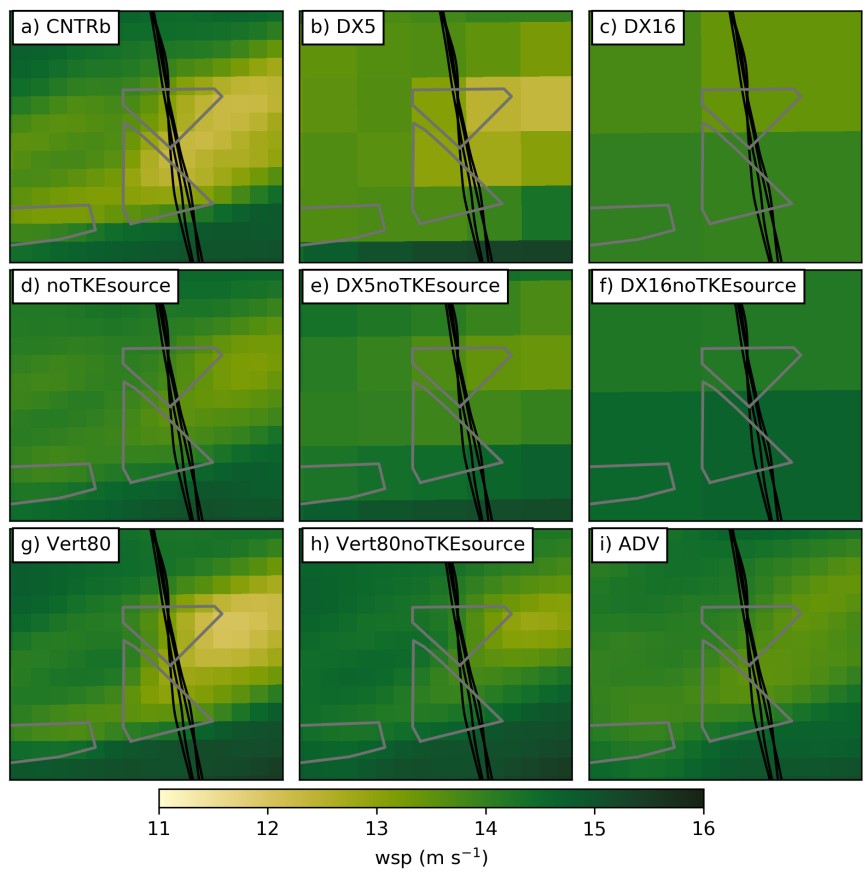

**Figure 11.** As in Fig. 7, but for the sensitivity simulations (b-i) DX5, DX16, noTKEsource, DX5noTKEsource, DX16noTKEsource at 15:00 UTC. For better comparison the control simulation CNTR is shown in (a).

Coarsening the horizontal resolution of the simulations to 5 km resolution degrades the agreement between the simulations and observations. As expected, the sharp drop in the horizontal wind speed in the observations at the southern edge of the wind farm oriented parallel to the incoming flow can not be represented in a mesoscale simulation with a horizontal grid size of 5 km, a result similar to the CNTRb simulation (Fig. 12b). However, the wake impact on the horizontal wind speed at 200 m AMSL (i.e. 60 m above the wind farms) is captured well, rooted in a TKE only 0.3 $m^2$ $s^{-2}$ lower than the observed mean (Fig. 12a, Fig. 13b), except for the region of strong horizontal shear that can not be captured by a mesoscale model.

Simulation DX16 reveals that a grid size of 16 km cannot capture the effect of wind farms with a size in the order of 100 $km^2$ by the use of a WFP. Compared to CNTRb the decrease in wind speed downwind of the wind farms GW is in the order of 1 m $s^{-1}$ instead of 2 m $s^{-1}$, suggesting that the vertical mixing is underestimated. Accordingly, Fig. 13c) reveals that the simulated TKE is two times lower than observed.





More vertical levels cause the same amount of TKE over the wind farm compared to CNTRb, (Fig. 12a, Fig. 13g) but the wind speed deficit at the southern edge of wind farm is in better agreement with the observations (Fig. 11g, Fig. 12b) by up to 0.5 m s$^{-1}$. Additionally, the wakes as seen in the SAR image match better with the ones simulated in Vert80 than in CNTRb (Fig. 4). The wakes in Vert80 (Fig. 4c) are more pronounced compared to CNTRb (Fig. 4b) and, hence, fitting better to the observed SAR image.

## 5.2 Sensitivity to vertical resolution with a disabled TKE source

For comparison to wind farm parameterizations without an explicit turbulence source (Volker et al., 2015), we conducted three simulations with the TKE source switched off using the CNTRb configuration and two coarser horizontal grids than in CNTRb (noTKEsource, DX5noTKEsource, DX16noTKEsource, Fig. 11d-f). Additionally, we performed a simulation having the TKE source disabled with 80 vertical levels, namely Vert80noTKEsource (Fig. 11h).

All simulations with the TKE source switched off show larger wind speeds over the wind farms. For example, in the simulation noTKEsource, wind speeds are ≈14 m s$^{-1}$ above the wind farm (Fig. 11d) associated with a lower TKE (Fig. 12a) than in CNTRb. Consequently, we expected even higher wind speeds in DX5noTKEsource over the wind farms associated with the lower TKE that can be resolved in a simulation with a grid size of 5 km. Indeed, the wind speed and the TKE over the farms is up to 0.5 m s$^{-1}$ and 0.3 m s$^{-1}$ lower than in CNTRb (Fig. 11e). Obviously, simulation DX16noTKEsource fails representing the impact of the wind farms on the wind speed (Fig. 11f) and the TKE (Fig. 13f).

Surprisingly, the simulation Vert80noTKEsource with 80 vertical levels shows approximately the same TKE as simulated in noTKEsource with 50 vertical levels, although more vertical shear should be resolved in Vert80 (Fig. 12a). This amount of TKE is similar to the noTKEsource simulation. Consistently, the wind speed reduction over the wind farm is almost similar to the noTKEsource simulation (Fig. 11b).

Summarized, all simulations without a TKE source produced too small of TKE compared to the observations. Therefore, we conclude that additional TKE is necessary to parameterize wind farms in mesoscale models in stable conditions.

## 5.3 Sensitivity to advection of TKE

The TKE advection option results in a greatly reduced TKE over the wind farm associated with a lower wind speed reduction over the wind farm. The simulated TKE is almost the lowest parameterized in all simulations performed for this study (Fig. 12a), resulting in an underestimation of the wind farm impact on the wind speed over the wind farm - 2 m s$^{-1}$ less than the observed mean deficit (Fig. 11i, Fig. 12b). However, the ADV (i.e. advection of TKE is active) simulation shows the highest TKE values within the wake of the wind farm GW (Fig. 13i).

## 5.4 Sensitivity to thrust coefficient

Two simulations (ThrustMin, ThrustMax) were performed to investigate the uncertainty introduced by the estimated thrust- and power coefficients. The corresponding uncertainty is shown in Fig. 12 as shaded area around the results of the CNTRb



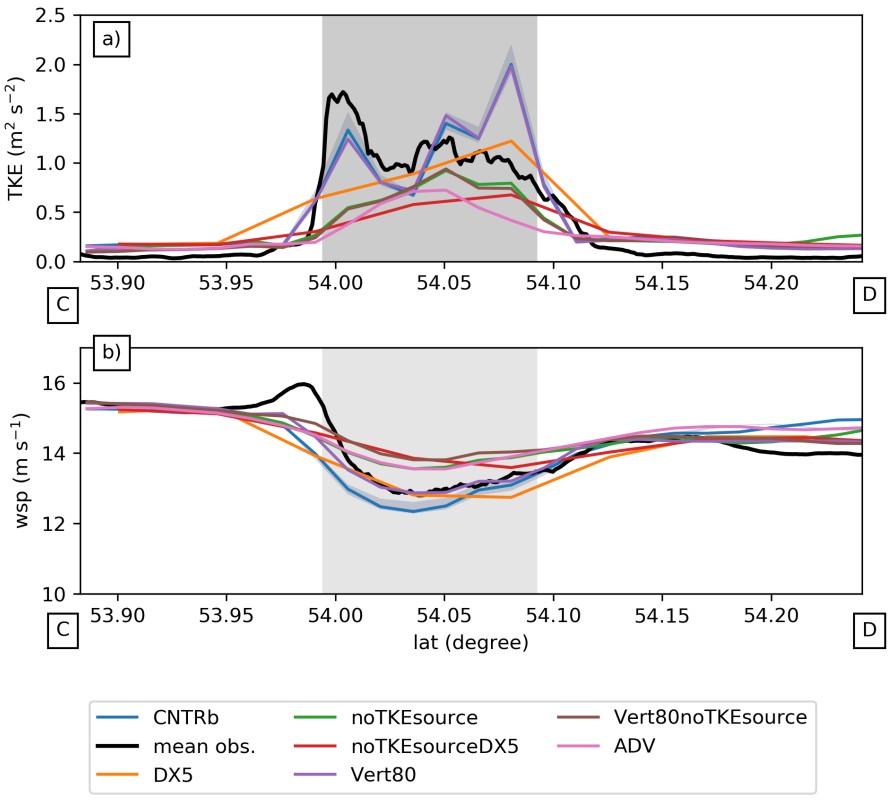

**Figure 12.** As in Fig. 8, but for TKE (a) and wind speed (b) for the sensitivity simulations DX5, noTKEsource, DX5noTKEsource, Vert80, Vert80noTKEsource and ADV conducted for case study II. For better comparison the control simulation CNTRb plotted as well.

simulation. The uncertainty in TKE due to the unknown power and thrust coefficients is smaller than the deviation caused by all sensitivity studies, except for simulation Vert80 (Fig. 12a). The uncertainty resulting for the wind speed deficit is smaller than the effect of all the other physics and numeric permutations tested here, including the effect of the vertical 80 levels (Fig. 12b).

## 6  Discussion

5   Obviously, the most important ingredient for simulating realistic wind speeds over offshore wind farms is the correct representation of the atmospheric state, regardless in which configuration the WFP is used. In two of the three case studies examined here, the simulations analyzed here failed to represent the atmospheric background correctly. WRF captured the state of the atmosphere for case II, as the boundary layer upwind of the wind farms GW was not modified by land. In contrast, the upwind conditions were not captured for the case studies I and III. Both cases were characterized by a large scale flow modified by

10   the land upwind. The model evaluation in section 4.1 revealed that the associated inversion in III that developed as warm air



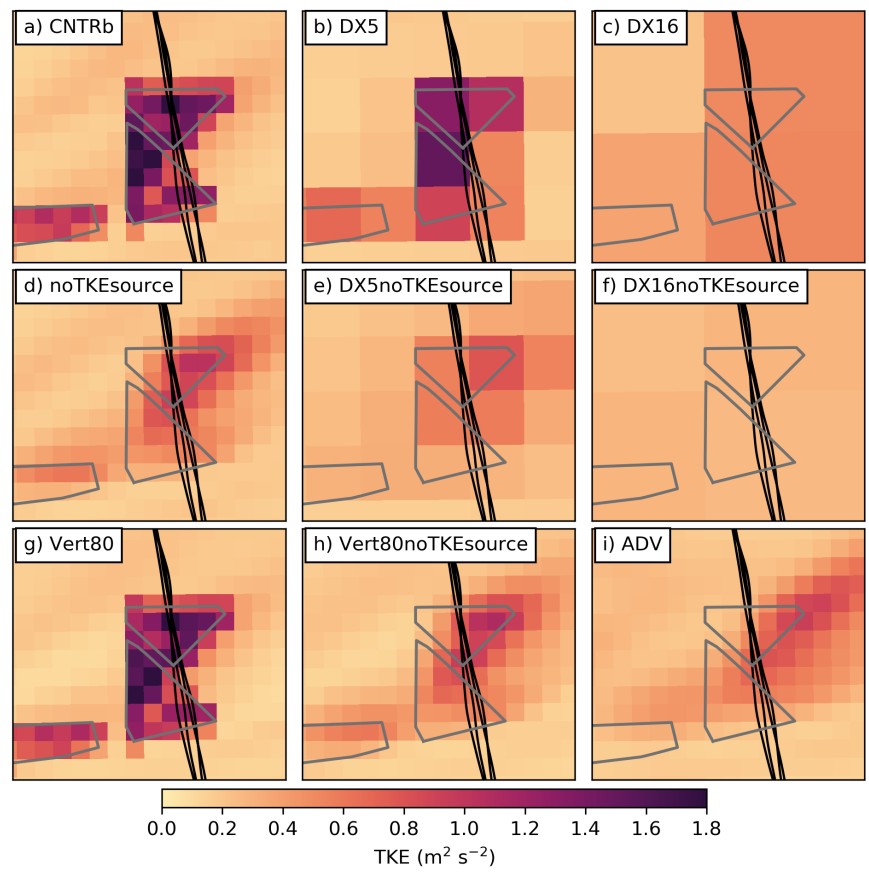

**Figure 13.** As in Fig. 9, but for the sensitivity simulations (b-i) DX5, DX16, noTKEsource, DX5noTKEsource, DX16noTKEsource, Vert80, Vert80noTKEsource, ADV at 15:00 UTC. For better comparison the control simulation CNTRb is shown in (a).

masses were advected from the land upwind over the German Bight, which is challenging for the simulation as described in Siedersleben et al. (2018b). The inversion almost decoupled the layer at 250 m from any processes below 200 m (i.e. top of inversion height), resulting in a very low signal of the wind farm in the TKE and wind speed (Fig. 9c, Fig. 10c). In contrast, the simulation showed TKE values up to 2.0 m$^2$ s$^{-2}$. However, during the additional two flyovers the TKE increased up to 2.0 m$^2$ s$^{-2}$ in the observations associated with a destabilizing MABL as Fig. 6c) reveals. Consequently, the TKE over the wind farms increased corresponding to an increased wind speed deficit during the last flyover. As the destabilization of MABL resulted in a profile with a potential temperature gradient similar to the simulated profile, the simulated and observed TKE have the same magnitude for the last two flyovers, underlining that the upwind conditions are crucial for representing the impact of offshore wind farms.





Our results suggest that, under stable conditions mesoscale wind farm simulations should use an additional TKE source, as the mixing and the associated wind deficit over the wind farms is too low otherwise.

Given the success with the TKE source switched on, we recommend for regional climate simulations using horizontal grids in the order of 5 km to use a WFP with an active TKE source. Additionally, the grid size must accommodate size of the wind

farms installed in the region of interest. For example, regional climate simulation using a grid as Vautard et al. (2014); Miller and Keith (2018) (i.e. 50 km, 30 km grid, respectively) would be unsuited for determining the climate impact of offshore wind farms on the German Bight, because we have shown that simulations with a horizontal grid size of 16 km are already too coarse to represent the impact on the MABL realistically. In contrast, simulations with a horizontal grid size of 5 km performed adequately when TKE was actively added to the model.

Strong shear lines at the edge of a wind farm or wake cannot be captured by mesoscale models. The strong horizontal shear observed at the wind farm GW at the southern edge oriented parallel to the impinging flow, has a horizontal extent of ≈2 km. Following Skamarock (2004), realistic solutions only exist for processes having seven times the grid size. Consequently, the horizontal shear with the associated TKE can not be represented by mesoscale models. However, in the simulation for case study II one could think that the model is able to present the shear line at the southern edge, when considering only the TKE.

Both the simulation and the observation show a TKE maximum at the southern edge of the wind farm. However, the peak in the observations was associated with the horizontal wind shear. In model the wind farms GW extend more to the south than in reality, hence, the TKE peak in the simulations is associated with the TKE source of the WFP and not with the horizontal shear.

The WFP's TKE source possibly introduces too much TKE on the upwind edge of a wind farm. Although the simulations did

not capture the atmospheric background in case study I, we noticed an important difference between the simulated and observed TKE. In the observations, the TKE over the wind farms increased as the flow penetrated through the wind farms (Fig. 10a), whereas the model adds the most TKE at the upwind side of the wind farm. The amount of TKE added to the model depends on the wind speed, number of wind turbines, and $C_{TKE}$ (see Eq. 1). Therefore, the added TKE is highest at the locations with the highest wind speeds within the farm, that is, at the front row of the wind farm. Of course, if the front row is associated

with a high wind turbine density, the WFP also adds the most TKE at the upwind side of the wind farm. In case study I, wind turbine density is high with up to five turbines per grid cell at the western edge of the wind farms (Fig. 2a). Additionally, we had south-westerly winds exposing the western edge of the wind farms to the highest wind speeds (Fig. 7a). Consequently, the simulated TKE has a maximum at the upwind side that was not observed in the aircraft measurements (Fig. 9a, Fig. 10a). However, without a TKE source, the deceleration was too low compared to the observations, especially when horizontal grids

are larger than or equal to 5 km.

The uncertainty of our simulations for case study II, introduced by the estimated thrust- and power coefficients is smaller than the effect of changing either the horizontal resolution or disabling the TKE source of the WFP of Fitch et al. (2012). Therefore, our sensitivity experiments conducted for case study II give useful and general recommendation for offshore wind farm simulations under stable conditions.





# 7 Conclusions

Using airborne measurements of wind speed and turbulent kinetic energy near offshore wind farms, we evaluate the wind speed and turbulent nature of the wind farm wakes as well as the parameterization of those wind farm wake effects enabled by the Weather Research & Forecasting Model (WRF) Wind farm parameterization (WFP) of Fitch et al. (2012). Our study

considered three cases at two different sites. Three case studies were, all characterized by stable conditions. During two case studies the marine boundary layer was highly influenced by the land upwind, resulting in deviations between observation and simulations. However, during one case study the impinging flow was coming from the west resulting in an inflow unaffected by any land. Hence, the WRF model represented the state of the atmosphere in the vicinity of the wind farms reasonable well. That allowed us to perform sensitivity studies in terms of horizontal and vertical resolution. Additionally, we investigated the effect

of the TKE source of the WFP of Fitch et al. (2012) on the MABL as well as the option of advecting TKE in the boundary layer scheme of WRF. These are our main findings:

– We recommend using the TKE source of the WFP of Fitch et al. (2012) for offshore wind farm simulations under stable conditions, especially for simulations having a horizontal grid coarser or equal to 5 km. However, we notice that the WFP adds too much TKE at the upwind side of a wind farm. We observed during two case studies that the TKE over the

wind farms increased with the path of the air through the wind farm, meaning that the TKE is higher at the downwind side of a wind farm than on the upwind side. In contrast, the WFP simulated the highest TKE at the upwind side of the wind farm associated with the highest wind speeds and wind farm density at the front row turbines. Nevertheless, the wind speed deficit is underestimated with disabled TKE source. Therefore, we suggest using the TKE source for stable conditions.

– Simulations using the WFP of Fitch et al. (2012) using a grid size of ≈15 km or more underestimate the impact of wind farms on the MABL under stable conditions, regardless of the mode of the TKE source. Given the fact that the impact of offshore wind farms is largest during stable conditions, we suggest that climate simulations assessing the impact of offshore wind farms should use a horizontal grid in the order of 5 km or finer. This horizontal resolution is difficult to achieve for global simulations, but feasible for regional climate simulations.

– In terms of the vertical resolution we obtained best results with 80 vertical levels, equal to a spacing of 12 m below 400 m AMSL as in Lee and Lundquist (2017). We tested two sets of vertical levels resulting in 3(1) and 13(4) levels below and within the rotor area. In case of an activated TKE source only minor differences were observed between the two sets of vertical levels. However, the wind speed deficit was captured better with the finer vertical resolution. Additionally, the simulated wakes agreed better with SAR data due to the smaller spacing of the vertical levels close to

the surface. Therefore, we recommend a spacing of the vertical levels in the order of ≈12 m for offshore simulations. In case computational resources are limited, simulations with a horizontal and vertical resolution of 5 km and 35 m below 100 m also captured the most important features over the wind farms.



- – Activating the TKE advection in the boundary layer scheme was associated with too low TKE over the wind farms that in turn resulted in an underestimation of the wind speed deficit over the wind farm.

These results support the hypothesis that the TKE source in the WFP of Fitch et al. (2012) is necessary under stable conditions at offshore wind farm sites, although we suggest that the added TKE is overestimated at the upwind side of the wind

farms, suggesting possible future improvements. Given the results of this study, previously published studies assessing the impact of offshore wind farms have possibly underestimated the impact on the marine boundary layer, hence, we suggest regional climate simulations for offshore sites with a grid size of 5 km or finer. However, the skill of such regional climate simulations is lessened when flow is from onshore due to difficulty parameterizing coastal effects. Thus future work should primary focus on boundary layer parameterizations that are able to capture the transition from land to open sea and vice versa.

*Code availability.* The WRF code is publicly available at http://www2.mmm.ucar.edu/wrf/users/downloads.html. The WRF configuration files and the wind farm parameterization of Fitch et al. (2012) with the TKE source switched off are available on GitHub (https://github.com/siederslebenEtAl2019/GMD2019)

*Data availability.* The airborne data is available on PANGAEA (https://doi.pangaea.de/10.1594/PANGAEA.902845) (Bärfuss et al., 2019). The location of the wind turbines are available at https://www.bundesnetzagentur.de. The ERA5 data that was used driving the WRF model

can be downloaded at https://cds.climate.copernicus.eu. The SAR data presented in this study can be accessed via https://scihub.copernicus.eu.

*Author contributions.* SKS and JKL outlined the manuscript and SKS wrote the manuscript with comments from JKL, AL, AP, BD, JSS, SE, JB and BC. BD drafted section 2.2. SKS and AP performed the analysis of the simulations and the aircraft measurements, respectively. AL, KB, AP and JB executed the aircraft measurements. AP and JB designed the flight pattern over the wind farms. SE and JKL supervised

SKS. The project WIPAFF was led by SE and TN.

*Competing interests.* The authors declare no competing interest

*Acknowledgements.* The WIPAFF project is funded by the German Federal Ministry of Economic Affairs and Energy (grant number: FKZ 0325783) on the basis of a decision by the German Bundestag.

JKL's effort was supported by an APUP agreement with the National Renewable Energy Laboratory. This work was authored in part by

the National Renewable Energy Laboratory, operated by Alliance for Sustainable Energy, LLC, for the U.S. Department of Energy (DOE) under Contract No. DE-AC36-08GO28308. Funding provided by the U.S. Department of Energy Office of Energy Efficiency and Renewable





Energy Wind Energy Technologies Office. The views expressed in the article do not necessarily represent the views of the DOE or the U.S. Government. The U.S. Government retains and the publisher, by accepting the article for publication, acknowledges that the U.S. Government retains a nonexclusive, paid-up, irrevocable, worldwide license to publish or reproduce the published form of this work, or allow others to do so, for U.S. Government purposes.

5    The Graduate School GRACE funded the research stay at CU of SKS. The WRF output was post processed by the use of the python library wrf-python (Ladwig, 2019) and visualized with matplotlib (Hunter, 2007). The simulations were performed on the keal cluster hosted by the IMK-IFU and gratefully managed by Benjamin Fersch. We would like to thank the aircraft crew Rolf Hankers, Thomas Feuerle, Helmut Schulz and Mark Bitter for conducting the flights. We visualized the WRF data by using colors as suggested by Stauffer et al. (2015) and Thyng et al. (2016). We also thank the European Space Agency (ESA) for making Copernicus Sentinel-1 SAR data freely available.



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
