# Peer review of "Turbulent kinetic energy over large offshore wind farms observed and simulated by the mesoscale model WRF (3.8.1)"

_Geoscientific Model Development, 2019_

## Referee Comment (RC1) · Anonymous Referee #1 · 29 Jul 2019

**Reviewers Comments**

**Title:** Observed and simulated turbulent kinetic energy (WRF 3.8.1) over large offshore wind farms

**Authors:** Simon K. Siedersleben, Andreas Platis, Julie K. Lundquist, Bughsin Djath, Astrid Lampert, Konrad Bärfuss, Beatriz Canadillas, Johannes Schultz-Stellenfleth, Jens Bange, Tom Neumann, and Stefan Emeis

**MS No.:** gmd-2019-100

**MS Type**: Model evaluation paper

**General comment**

This paper describes observed and simulated wind speed and TKE over offshore wind farms in the North Sea. Three case studies are analysed by means of airborne measurements and real case WRF simulations to investigate the impact of offshore wind farms on the boundary layer flow and the generation of TKE. The observations are used to verify a large number of sensitivity tests with focus on the TKE production of the wind farm parameterization (WFP) available in WRF.

The paper is well written and the test site, the observations and the numerical setup are described in a clear and understandable way. The analysis of the three cases demonstrates the importance to simulate the upstream boundary layer correctly, which was not the possible in the cases I and III probably due to the interaction between land and sea surfaces. The study is very helpful to optimize the use of the WFP in numerical simulations and shows possible room for improvement of this parameterization. I don't have any major comments and recommend minor revision for the submitted manuscript.

**Minor comments**

1. Title: Is it necessary to show the WRF versin in the caption (WRF 3.8.1)? I would let it out in the title.
2. P1L11: I think „deficit" is missing and it should be: „... which in turn causes an underestimation of the wind speed deficit above the wind farm".
3. P2L8: „...the wind speed reduction caused by the wind turbines upwind can be only balanced by te vertical momentum flux." I would omit this sentence, as it repeats the info given in the sentence before.
4. P2L22-L24: I think the two sentences are a little bit confusing. Can you rewrite them, maybe something like: „It is therefore necessary to evaluate TKE of mesoscale wind farm parameterizations with observed TKE over large offshore wind farms."

5. P2L29: Are you only interested in TKE above wind farms or also in TKE behind them (downstream in the wake)?

6. P3L2: I would omit „version 3.8.1" and add this info to section 2.3.

7. P3L5: „horizontal and vertical grid resolution"

8. P3L9 and P3L25: I think an article is missing: ...in the vicinity of the…

9. P5L9: Please add: Figure 4a) shows the 10m wind speed …

10. P9 Table2: I don't understand the differences of the three control simulations CNTRa, CNTRb, CNTRc. Is it right that the setup of these runs is the same and they only differerence is the case study (I, II and III). For me it's a little bit confusing to have these three control simulations and I would suggest to have just one CNTR setup in Table 2.

11. P10L4: I don't understand the explanation of „Warm air advection was associated with a stably stratified atmosphere according ..." Normally, warm air advection is associated with an anticyclonic turning of the geostrophic wind with height (on the northern hemisphere turning to the right). Can you add one sentence here to explain in more detail where you can see  that warm air advection occured?

12. P10L6: Can you explain where the FINO1 tower is located or add it maybe in Fig. 1?

13. P11L13: Can you replace „the airborne measured TKE" by „the observed TKE"?

14. P11L13: The sentence „The TKE over the wind farms MSO and ONO..." is a little bit long and difficult do understand. Can you please simplify this sentence?

15. P13L13: I don't understand the explanation with the warm air advection. My explanation for the slight disagreement between WRF and the observation is that for case study I we are close to an approaching trough advecting cold air from northwest. It might be that the location of the trough is slightly shifted in the model and that we are located already in colder and different airmasses compared to the observations. This is, however, just a guess...

16. P17L11: Fig. 11 should be mentioned in the text before Fig. 12. You could add a hint to Fig. 11 the sentence „A summary of all sensitivity tests...".

17. P18L1: Please add (see Fig. 12) at the end of the first sentence.

18. P20L3: Replace the number 80: „...the effect of the 80 vertical levels..."

19. P24L5: Please simplify the sentence „Given the results of this study, ...", as it is difficult to read.

20. P24L8: „... difficulty in  parameterizing..."

**Figure comments**

1. Fig. 2: Please make a link to Table 2 in the caption in line 5: … for the sensitivity studies: DX5, DX16, … (see Table 2).

2. Fig. 8 and Fig. 10: I'm wondering, if it is possible to add an arrow in each panel, which indicates the mean wind direction along each flight leg. In leg AB the mean wind along the cross section is blowing from B to A, in leg CD from C to D and in leg EF from F to E (please correct me if this is not right). For leg CD it's maybe difficult as the leg seems to be nearly perpendicular to the approaching wind. I think such arrows could help to identify up- and downwind region of the wind farms. Anyway this is just a suggestion...

3. Fig. 7, 9, 11, 13: Is it possible to add the letters A, B, C, D, E, F which label the cross sections? I know that they are in Fig. 1, but it would help to see at a glance how the legs were oriented?

---

## Referee Comment (RC2) · Anonymous Referee #2 · 10 Sep 2019

General Comment: The manuscript by Siedersleben et al provides useful and interesting results about large scale wind farm cluster wakes and the modelling of their impact by means of mesoscale simulations. The focus on comparing turbulent kinetic energy and its sensitivity in the modelling is an important and actual research topic and has not been studied extensively. In general, the manuscript is also quite well written and the figures are easy to read. However, I have four major and a number of minor comments and thus recommend the publication of the manuscript after dealing with major revisions.

Major Points: 1) The reviewer had experienced problems with the data of the Bundesnetzagentur (coordinates and information such as hub heights and diameters) and knows that these data need to be taken into account with great caution. For instance, the wind farm Meerwind Südost consist of 80 turbines according to the operator [1]. The authors write that only 74 turbines were accounted for! The authors will thus should double check the coordinates and all other information that were taken from the Bundesnetzagentur data and correct all the wrong information and rerun all cases that used wrong specifications.

2) A quite big difference between the surface measurements at FINO1 and the air temperature above is reported in two cases that does not physically fit to the measured stability above the sea. The reviewer suggests to compare several other sea surface stations (e.g. BSH buoys) in the surroundings and also a high resolution sea surface dataset such as OSTIA to verify the measurements at FINO1. These could also be affected by algae growth and thus have a bias.

3) When measuring wakes it is important to know the operational state of the wind farms that are responsible for the wakes. This data is mostly confidential. However, public data sources (such as e.g. the Energy charts [2]) allow for estimating if the energy production at the respective wind speed is realistic and the majority of the turbines were actually operating. This should be done by the authors for the cases presented here.

4) Between WRF 3.7.X and WRF 4.X a change of a constant in the MYNN model (COARE OPT from 3.0 to 3.5) led to significant differences and biases in the wind (see e.g. [3]). Thus, this parameter was changed back later by the model developers (see e.g. [4] and [5]) in the WRF standard version. The reviewer suggests to test the impact of this on the results by comparing a different WRF version or changing it manually in the version used by the authors.

Minor Points:

Title: The reviewer suggest to rewrite the title to something like: Turbulent kinetic energy over large wind farms observed and simulated by the mesoscale model WRF

Abstract: Because. . .. -> You should not start a sentence with . . . because. . ...

Page 1 – Line 15: Offshore wind energy. . .. -> the reviewer suggest to also give a time frame since when. Maybe something like "in recent years" or "in the last decace"

Page 2 – Line 9: Given generally low mean vertical velocities -> Yes, this is of course true but only for the ambient flow. The vertical velocities close to the rotor can be very high due to the rotational speed. Thus, either you should add the scales you are looking at (on mesoscales) or add the information that this is about the ambient flow.

Page 2 – Line 11: Global and regional climate simulations . . .. -> This term is used a few times in the paper but WRF is in its first place a weather model. Thus, the reviewer suggest to add "weather models" here.

Page 2 – Line 17: . . . both WFPs delivers. . . -> deliver (no s)

Page 2 – Line 28: . . . we want to know -> the reviewer suggests to rewrite: ". . . we try to find answers to the following research questions."

Page 3 – Line 8: Unique. . .. -> Do you mean "only"?

Page 3 – Line 9-10: As we have. . . -> This sentence is difficult to understand. Maybe rewrite: We have a SAR satellite image for one of the cases investigated in this study, a brief. . ..

Page 3 – Line 13: It would be good to have an introductional sentence here, something like: In the framework of the WIPAFF project, the research aircraft Dornier 128-6 operated by TU Braunschweig was used for measuring the wakes of several large offshore wind farms within several campaigns. In these campaigns, the aircraft flew with a true air speed of.....

Page 3 – Line 15: . . . have a resolution of 0.66m -> do you mean .. provide values ever 0,66m of a flight leg?....

[Figure]

Page 3 – Line 24: . . . have the same pattern. -> Suggestion: . . . were conducted using the same pattern.

Page 4 – Figuer 1 (Caption): . . . are in use. -> in operation

Page 4 – Figure 1 (Caption): . . . in 2017 -> I suggest to rewrite: ". . . during the time period investigated in this study.

Page 5 – Line 2: . . . such as Synthetic Aperture Radar (SAR) . . . -> I suggest to add: 'sattelite-borne Synthetic Aperture Radar'

Page 5 – Line 8: . . . repeat cycle of 6 days, -> Do you mean return frequency/period?

Page 5 – Line 9: . . . Sentinel 1 passes in the German bight. . . -> This sentence is hard to understand. I suggest something like: Sentinel 1 passes the German Bight at 5am or 5pm

Page 5 – Line 17: 16 km in the outermost domain followed by two domains with 5 km and 1.67 km,. . . -> Usally a factor of 3 or 5 is used in WRF simulations. Do you mean 5.3 km? (if yes, please correct throughout the paper)

Page 5 Line 20-22: The authors reference the study by Hahmann which shows that a spin-up time of more than 12 hours should be used and then don't do this in the third case. Why? If there is no good reason for this, the third case should be re-run with a spinup-time of 12 hours or larger.

Page 7 – Figure 4: It is quite difficult to distinguish wind farms and islands especially close to the coast (e.g. Borkum Riffgat). The reviewer suggests to change the wind farm shapes to some greyish color.

Page 8 – Line 1: A summary of our sensitivity tests appears in Table 2. –> The reviewer suggests to rewrite: A summary of the setups of our sensitivity tests appears in Table 2.

Page 9 – Table 3: Meerwind Südost consists of 80 turbines (see Major Point 1). Also,

the reviewer suggests to double check all other information (hub heights/diameters) with the official information from the wind farm operators or turbine manufacturers!

Page 10 – Line 1: "stably stratified" –> The reviewer suggests to rewrite: "slightly stably stratified" because the vertical change of the potential temperature with height over the rotor area is very small.

Page 10 – Line 5 (Fig 1 – black thich line). It took me really long to find the black thick line. Maybe add a marker to the line? Or Markers at start and end? (also for the other lines)

Page 10 –Line 6-7: approx. 2 K warmer than the air temperature –> This is really a lot and thus should be explained. The sensors at FINO1 might be impacted by algae growths etc. You should check with other BSH buoys in the German Bight and a high-resolution sea surface dataset e.g. the OSTIA data if this measurement at FINO1 is really valid. This is also true for case III (see below).

Page 10 – Line 14-15:Further to the north. . . -> This is true but Sandbank is also very "thin" for westerly winds (only 2 rows of turbines). This point should be added to the discussion.

Page 10 – Line 21-22: However an SST of 288.5 K was measured at FINO1. . .. (see second last comment above). If it holds to be correct, a discussion should be added that the stability could also be an advective feature coming from another location.

Page 10 Line 25: Wind speed over. . .. –> The reviewer suggests to correct to "above". This occurs several times in the document.

Page 11 – Line 3-4: We suggest that this acceleration emerges due to an enhanced flow around the wind farm due to the stably stratified atmosphere –> This is of course correct but needs further explanation. The reviewer suggests to add something like: This stability hinders the flow to extend vertically above the wind farm and forces it to flow around the wind farm leading to a speed up at the wind farm edges.

Page 11 – Line 12: TKE over and next to the wind farm -> Suggestion: TKE above and next to the wind farm

Page 11 – Line 18: . . . where the shear was greatest (shown in Fig. 8a) –> The reviewer suggests to change to "largest"

Page 12 – Line 6: . . . where the horizontal wind shear was greatest –>The reviewer suggests to change to "largest"

Page 13 – Line 3: In every case -> The reviewer suggests to change to "In summary, in every case . . .."

Page 13 – Line 21: "interpolating the wind speed to a height of 10 m is difficult." –> The WRF model provides the output of U10/V10 which are of course not from a model level but should give comparable results to the 10m wind speed. The authors thus should use these data

Page 13 – Line 28: "of the atmosphere in the model did not change with time." –> better: over time

Page 14 – Line 2: "at 200 m AMSL over and next to the wind farms" –> better: above and next to the wind farm

Page 14 – Line 4: "A weak increase in wind speed similar. . ." –> What this the physical explanation for this increase?

Page 15 – Figure 8 (also 10,12) –> The reviewer suggests to change the x- axis to kilometres from wind farm center and define a point in the center of the wind farm. This way it is very difficult to estimate what the extension of effects such as the speed up are.

Page 15 – Line 6: "showed an increased impact on the wind speed" –> Do you mean "showed an increase of the wind speed above the wind farm"?

Page 16 – Line 5: "for the simulations conducted for 14 October 2017" –> The reviewer

suggests: "for the simulations of 14 October 2017".

Page 17 – Line 6: It is very confusing that the CNTRns are not related to the cases here. The reviewer suggests to add this where needed. E.g. change to 'CNTRb – case II'

Page 17 – Line 10: "active" –> do you mean activated?

Page 19 – Line 6: "resolution with a disabled TKE source" –> The reviewer suggests to rewrite to "without a TKE source" to make it more general.

Page 19 – Line 7: "without an explicit turbulence source (Volker et al., 2015)" –> I think it would be really nice and strengthen the work if another case simulated with the model by Volker would be added but I also see that this would probably be a lot of additional work. . ... So maybe add this to the outlook at the end of the paper?

Page 19 – Line 21: ". . .produced too small of TKE compared. . .." –> Do you mean ". . .. produced too small amounts of TKE compared. . ."?

Page 19 – Line 24: "in a greatly reduced TKE" -> The reviewer suggests to write "largely reduced TKE".

Page 19 – Line 26: "over the wind farm - 2 m s$-1$" -> Better: "over the wind farm of – 2 ms-1"

Page 22 – Line 10: "Strong shear lines at the edge of a wind farm or wake. . ." -> The reviewer suggests to add: that originate in the flow around the wind farm.

Page 22 –Line 16: "In model the wind farms GW" –> In the model the wind farms GW

Page 23 – Line 24: "but feasible for regional climate simulations." -> The reviewer suggests to add "weather simulations"

Page 24 – Line 8: The reviewer suggests to change this last sentence of the Conclusions: Thus, future work on wake effects of large offshore wind farms should primarily

focus on. . ...

Page 28 – Line 23-24: Svensson et al –> doi missing

References:

[1] https://www.windmw.de/meerwind.html

[2] https://www.energy-charts.de/

[3] https://zenodo.org/record/2682604

[4] https://github.com/NCAR/WRFV3/blob/master/phys/module_sf_mynn.F

[5] https://github.com/wrf-model/WRF/blob/master/phys/module_sf_mynn.F

---

## Short Comment (SC1) · 10 Sep 2019

Thank you for the efforts you have gone to to make this science reproducible. Currently the code and data availability falls just short of required GMD standards, but in ways that should be very straightforward to remedy in the revised submission.

Code availability

GitHub is an excellent development resource, but it is not a suitable permanent archive location. Indeed, GitHub themselves tell you to use Zenodo for this purpose, and

provide a very easy mechanism to archive your GitHub repository on Zenodo. See https://guides.github.com/activities/citable-code/. Please use the facility there and cite the resulting DOI. You may, of course, refer to the GitHub repository in addition to the DOI if you wish (though the Zenodo repository will also link to it).

Data availability

ERA5 is a big dataset! You do not seem to refer to exactly which data is used and therefore should be downloaded. Please do so. If it's too cumbersome to put it directly in the paper, you could, for example, provide the details in your model configuration repository.

For full details of current GMD code and data requirements, please see: https://www.geoscientific-model-development.net/about/code_and_data_policy.html

---

## Referee Comment (RC3) · Tobias Ahsbahs (Referee) · 19 Sep 2019

**Summary**

The authors of the manuscript studied the atmospheric boundary layer above offshore wind farms for three cases. Co-located aircraft measurements are available and horizontal wind speed and TKE are measured. In particular their sensitivity study of different setups regarding the horizontal and vertical resolution, influence of turbulence source terms, and turbulence advection. Having high resolution reference measurements makes this work useful as it can go beyond stating differences between model setups. The manuscript is well written and concise. I recommend publication but have

some minor points that could make the manuscript easier to follow for the reader.

**General comments**

- It is hard to follow where the wind is coming from in the plots. The flights are neither cross wind nor along the wind direction but at an angle. This needs to be clarified earlier as it took me until page 14 to understand this. E.g. Fig 8: It is not really clear where the wake is.

- Synoptic forcing is mentioned but not clearly defined where it is taken from.

- The naming of the simulations does not make it clear which cases have been simulated. I would presume that all cases would be simulated with all the simulation setups in Tab 2, which would correspondent to 99 different simulations. Has this been done?

- Please check references to Figures carefully, I noticed some that seemed to reference the wrong Figure but might have not caught them all.

- Please increase the size of Fig. 4 and 5. They greatly helpful understanding the paper but are hard to read.

- In the Discussion you give recommendations that can be understood as quite general. They are based on one case, where WRF got the background flow correctly. Please stress this fact more as it poses a clear limitation. A reader who scans the Abstract and Discussion would benefit from this.

**Specific comments**

- P3, L15 - 17: This sentence is confusing and I am not sure what is meant.

- P5 l33 to P6 L1: The thrust coefficient is related to the thrust of the wind turbine not the energy.

- Fig. 2. Simulations are mentioned (CNTR) but not introduced beforehand. This might confuse the reader.

- Eq. 1. In the text at P7 L 4 to 7 you explain what $V_H$ and $V_{ij}$ are. How is $V_H$ different from $V_{ij}$?

- Fig. 3: Similar issue as before. It is unclear what the simulation setups are at this point.

- P9, Tab 2: Add which cases are simulated. What is the difference between CNTRa, CNTRb, and CNTRc?

- P10, L1-7: How was stratification determined here? The measured wind speed profile does not suggest much shear and the potential temperature profiles shows very little variation.

  Some problems with Figure references (6g is wind direction but temperature is referred). Fig. 1 is referenced but probably Figure 6 meant.

- P10ff: Which simulation from Tab. 1 are presented here? What resolution?

- Fig. 7: It is hard to make out differences in the wind speed with the chosen wind speed scaling. Please split it by case to make it easier for the reader.

- Fig. 8: It is not intuitive to the reader to follow if a wake effect is expected outside of the wind farm region. It would help to add another shade to where a wake effect would be expected. This is linked to the first general comment.

- P18, L1: Reference to Figure 12. L5: "TKE only $0.3 m^2 s^{-2}$ lower than the observed mean". Is this the mean over the wind farm location? L6: Which area with

high shear? Are the referenced Figures correct? L10: Within Figure 13 there is no measurement for easy comparison. It could be helpful to define an average TKE over the wind farm and present this consistently for all simulations and the observation.

- P19, L7: Is the EWP from Volker et.al.2015 used here or is the turbulence production term turned off in the Fitch parametrization? Please clarify.

––––––––––––––––––––––––––––––

---

## Author Response (AR1)

Answers to the comments of Referee #1, #2, #3 and the interactive comment of David Ham corresponding to the article "Observed and simulated turbulent kinetic energy (WRF 3.8.1) over large offshore wind farms"

First of all, we thank the referees for their comments, especially reviewer #2 helped to improve the manuscript. In this document the comments of the referees are plotted in black the corresponding answers are printed in green.

Reviewer #1

**Minor comments**

1. Title: Is it necessary to show the WRF versin in the caption (WRF 3.8.1)? I would let it out in the title.

   According to the guidelines of GMD it is necessary to state the version number in the title of the manuscript

2. P1L11: I think „deficit" is missing and it should be: „... which in turn causes an underestimation of the wind speed deficit above the wind farm".

   Great catch, thanks. (P1L11)

3. P2L8: „...the wind speed reduction caused by the wind turbines upwind can be only balanced by te vertical momentum flux." I would omit this sentence, as it repeats the info given in the sentence before.

   We agree, thanks.

4. P2L22-L24: I think the two sentences are a little bit confusing. Can you rewrite them, maybe something like: „It is therefore necessary to evaluate TKE of mesoscale wind farm parameterizations with observed TKE over large offshore wind farms."

   We agree and omit the last sentence of the paragraph and write:" However, an accurate representation of observed TKE and the associated change in the vertical fluxes over a wind farm is difficult to evaluate but necessary." (P2L21)

5. P2L29: Are you only interested in TKE above wind farms or also in TKE behind them?

   In this case, only above offshore wind farms as we only have measurements there. To be more precise, we could write therefore: "How sensitive is the impact of the wind farm parameterization on the TKE to the horizontal and vertical grid of the driving mesoscale model **above offshore wind farms**?" However, that makes difficult to understand. (P2L31)

6. P3L2: I would omit „version 3.8.1" and add this info to section 2.3.

   We agree.

7.  P3L5: „horizontal and vertical grid resolution"

Done. (P3L3)

8.  P3L9 and P3L25: I think an article is missing: ...in the vicinity of the...

Done. (P3L7 and P3L26)

9.  P5L9: Please add: Figure 4a) shows the 10m wind speed ...

Done. (P5L9-10)

10. P9 Table2: I don't understand the differences of the three control simulations CNTRa, CNTRb, CNTRc. Is it right that the setup of these runs is the same and they only differerence is the case study (I, II and III). For me it's a little bit confusing to have these three control simulations and I would suggest to have just one CNTR setup in Table 2.

Indeed, that can be confusing. We added an extra column to Table 2 naming the corresponding case study and added an extra paragraph: „We conducted three control simulations namely CNTRa, CNTRb and CNTRc corresponding to the three case studies. These three simulations are identical in terms of their numerical setup, they only simulate different days." (P5L20-24)

11. P10L4: I don't understand the explanation of „Warm air advection was associated with a stably stratified atmosphere according ..." Normally, warm air advection is associated with an anticyclonic turning of the geostrophic wind with height (on the northern hemisphere turning to the right). Can you add one sentence here to explain in more detail where you can see that warm air advection occured?

True. The wind is turning anticyclonic in Fig. 6g). However, the turning is not pronounced. Therefore we reformulated the sentence: "Weak warm air advection was associated with a slightly stable stratified atmosphere according to the climb flight (Fig 6g, anticyclonic turning of the wind with height)" (P10L11-12)

12. P10L6: Can you explain where the FINO1 tower is located or add it maybe in Fig. 1?

Sure. We have added the location of the tower in Fig. 1.

13. P11L13: Can you replace „the airborne measured TKE" by „the observed TKE"?

Done. (P13L1)

14. P11L13: The sentence „The TKE over the wind farms MSO and ONO..." is a little bit long and difficult do understand. Can you please simplify this sentence?

True, this sentence was too long. We shortened this sentence and write now: "The TKE over the wind farms MSO and ONO was increased compared to the surrounding (Fig. 9a, Fig. 10a). More specifically, the research aircraft measured a TKE of up to

2.0 m$^2$ s$^{-2}$, but 0.2 m$^2$ s$^{-2}$ within the undisturbed environment, meaning that the TKE over the wind farms is almost ten times higher 50~m over the rotor top compared to the surrounding environment. " (P13L1-5)

15. P13L13: I don't understand the explanation with the warm air advection. My explanation for the slight disagreement between WRF and the observation is that for case study I we are close to an approaching trough advecting cold air from northwest. It might be that the location of the trough is slightly shifted in the model and that we are located already in colder and different airmasses compared to the observations. This is, however, just a guess...

Yes, I think we both mean the same think: The deviation between simulation and observation stems from the advection of different air, that in turn is caused by different wind directions. We like your explanation and adapted it:" This deviation could be rooted in a dislocation of the incoming trough, causing more westerly winds in the simulations than in the observations." (P130-31)

16. P17L11: Fig. 11 should be mentioned in the text before Fig. 12. You could add a hint to Fig. 11 the sentence „A summary of all sensitivity tests...".

A very helpful comment. We rearranged the order of the figures and added the following line to the captions of Figures 11, 12 and 13:" A summary of all sensitivity test is listed in Table 2".

17. P18L1: Please add (see Fig. 12) at the end of the first sentence.

Done. (P17L10)

18. P20L3: Replace the number 80: „...the effect of the 80 vertical levels..."

Done. We write now: "including a change in the number of vertical levels." (P22L8)

19. P24L5: Please simplify the sentence „Given the results of this study, ...", as it is difficult to read.

True. We have split this sentence into two parts:" Given the results of this study, previously published studies assessing the impact of offshore wind farms have possibly underestimated the impact on the marine boundary layer. Hence, we suggest regional climate simulations for offshore sites with a grid size of 5~km or finer." (P25L1-3)

20. P24L8: „... difficulty in parameterizing..."

Done. P25L4

**Figure comments**

1.  Fig. 2: Please make a link to Table 2 in the caption in line 5: ... for the sensitivity studies: DX5, DX16, ... (see Table 2).

    Done.

2.  Fig. 8 and Fig. 10: I'm wondering, if it is possible to add an arrow in each panel, which indicates the mean wind direction along each flight leg. In leg AB the mean wind along the cross section is blowing from B to A, in leg CD from C to D and in leg EF from F to E (please correct me if this is not right). For leg CD it's maybe difficult as the leg seems to be nearly perpendicular to the approaching wind. I think such arrows could help to identify up- and downwind region of the wind farms. Anyway this is just a suggestion...

    There is a misunderstanding. In Fig. 8 and Fig. 10 the mean wind is more or less perpendicular to the cross sections i.e. the mean wind points into the paper plane as it is indicated in Fig. 8. and Fig. 9. To avoid any confusion here, we added a sentence to the caption of Fig. 8 and Fig. 9 stating that the flight legs are perpendicular to the mean wind speed. We apologize for this misunderstanding.

3.  Fig. 7, 9, 11, 13: Is it possible to add the letters A, B, C, D, E, F which label the cross sections? I know that they are in Fig. 1, but it would help to see at a glance how the legs were oriented?

    In principle that is possible. However, the close-ups shown in Fig. 7, 9, 11, 13 are so focused on the wind farms that the points A, B, C, D, E, and F are located outside of the Figure frame.

Reviewer #2
**General Comments**

1) The reviewer had experienced problems with the data of the Bundesnetzagentur (coordinates and information such as hub heights and diameters) and knows that these data need to be taken into account with great caution. For instance, the wind farm Meerwind Südost consist of 80 turbines according to the operator [1]. The authors write that only 74 turbines were accounted for! The authors will thus should double check the coordinates and all other information that were taken from the Bundesnetzagentur data and correct all the wrong information and rerun all cases that used wrong specifications.

First things first, we are impressed by the detailed knowledge of the reviewer. This was an important catch although it wasn't having any substantial impact on our results. A comparison of the old and new Figure 2 of the manuscript is shown in Figure 1; two grid cells were affected. Consequently, we also had to redo the simulations and Figures, especially for case study one, where we did measurements over the wind farm Meerwind Süd|Ost.

However, no changes in the TKE along the flight track could be notified as the flight track was located more to the north of the affected grid cells (Fig. 1b). Anyhow, the TKE as shown in Figure 9 in the manuscript, obviously changed as the number of turbines increased in two grid cells (see annotation in Fig. 2b).

Additionally, we double checked all other wind turbine data as pointed out by the reviewer.

[Figure]

*Figure 1: Number of wind turbines per grid cell in the innermost domain (i.e. grid size of 1.67 km) in the previous version (a) and corrected version (b). The grid cells affected by the corrections are labeled (b).*

2) A quite big difference between the surface measurements at FINO1 and the air temperature above is reported in two cases that does not physically fit to the measured stability above the sea. The reviewer suggests to compare several other sea surface stations (e.g. BSH buoys) in the surroundings and also a high resolution sea surface dataset such as OSTIA to verify the measurements at FINO1. These could also be affected by algae growth and thus have a bias.

Before we started this campaign we also had the idea that a stable stratification above the ocean is always associated with a colder SST than air temperature. However, during the study of the aircraft observation for the publication Siedersleben et al. (2018) we noticed several cases where we had a colder air temperature 40 m above mean sea level (AMSL) than the SST associated with a stable stratification at rotor height. For completeness the SST measured by the BSH buoy at FINO1 and the OSTIA SST surface temperature is shown in Fig. 2, confirming the SST of FION1.

[Figure]

*Figure 2: SST at FINO1 and according to the OSTIA data set. For a quick comparison the SST data was also compared to the OSTIA data set.*

3) When measuring wakes, it is important to know the operational state of the wind farms that are responsible for the wakes. This data is mostly confidential. However, public data sources (such as e.g. the Energy charts [2]) allow for estimating if the energy production at the respective wind speed is realistic and the majority of the turbines were actually operating. This should be done by the authors for the cases presented here.

We checked the "energy charts" as suggested by the reviewer. However, the wind farm Meerwind Sued Ost is sadly not available at the suggested homepage. We summarized the information below in three tables, whereby each table corresponds to one case study. According to the aircraft obervations case study II was characterized by the highest wind speeds (14 m s$^{-1}$) corresponding to the largest operating grades of all three case studies i.e., Godewind 2 shows an operational grade of 93 % during the observational period indicating that all wind turbines were operational. To clarify this point we added an extra paragraph in section 2.4. P10L2-4

**case study 1, 09 August 2017, 16:00-17:00 local standart time (LST)**

wind speed according to climb flight at hub height 11 m/s

| wind farm | power (GW) | nominal power (MW) | operating grade |
|---|---|---|---|
| Amrumbank West | 0,223 | 301 | 0,740863787 |
| Meerwind Süd Ost | ? | ? | ? |
| Nordsee Ost 1 | 0,107 | 144 | 0,743055556 |
| Nodsee Ost 2 | 0,119 | 144 | 0,826388889 |

**case study 2, 14 October 2017, 16:00-17:00 LST**

wind speed according to climb flight at hub height 13-14 m/s

| wind farm | power (GW) | nominal power (MW) | operating grade |
|---|---|---|---|
| Gode 1 | 0,294 | 332 | 0,885542169 |
| Gode 2 | 0,236 | 252 | 0,936507937 |
| Borkum Riffgrund 1 | 0,256 | 332 | 0,771084337 |
| Nordsee One | 0,217 | 312 | 0,695512821 |

**case study 3, 15 October 2017, 12:00-14:00 LST**

wind speed according to climb flight at hub height 10 m/s

| wind farm | power (GW) | nominal power (MW) | operating grade |
|---|---|---|---|
| Gode 1 | 0,196 | 332 | 0,590361446 |
| Gode 2 | 0,122 | 252 | 0,484126984 |
| B. Riffgrund 1 | 0,11 | 332 | 0,331325301 |
| Nordsee One | 0,145 | 312 | 0,46474359 |

4) Between WRF 3.7.X and WRF 4.X a change of a constant in the MYNN model (COARE OPT from 3.0 to 3.5) led to significant differences and biases in the wind (see e.g. [3]). Thus, this parameter was changed back later by the model developers (see e.g. [4] and [5]) in the WRF standard version. The reviewer suggests to test the impact of this on the results by comparing a different WRF version or changing it manually in the version used by the authors.

We are aware of that fact. We tested the setup of version WRF 3.5 for the simulations conducted in Siedersleben et al. (2018a) and could not see any improvement. Therefore, we used the model setup as in Siedersleben et al. (2018a) also for this publication.

**Minor comments**

Title: The reviewer suggest to rewrite the title to something like: Turbulent kinetic energy over large wind farms observed and simulated by the mesoscale model WRF

We adapted your suggestion: "Turbulent kinetic energy over large wind farms as observed and simulated by the mesoscale model WRF (3.8.1)"

Abstract: Because. . .. -> You should not start a sentence with . . . because. . ...

Ok. We write now: "Wind farms affect local weather and microclimates, hence, parameterizations of their effects have been developed for numerical weather prediction models." P1L1

Page 1 – Line 15: Offshore wind energy. . .. -> the reviewer suggest to also give a time frame since when. Maybe something like "in recent years" or "in the last decace"

Done: Offshore wind energy in Europe gains importance every year in the last decade. P1L15

Page 2 – Line 9: Given generally low mean vertical velocities -> Yes, this is of course true but only for the ambient flow. The vertical velocities close to the rotor can be very high due to the rotational speed. Thus, either you should add the scales you are looking at (on mesoscales) or add the information that this is about the ambient flow.

Thanks, good point. We added the scale we are looking at and write now: "Given generally low mean vertical velocities on the mesoscale, the turbulent vertical momentum flux is crucial when determining the power density of large offshore wind farms." P2L9

Page 2 – Line 11: Global and regional climate simulations . . .. -> This term is used a few times in the paper but WRF is in its first place a weather model. Thus, the reviewer suggest to add "weather models" here.

True. We write now: "Climate simulations and weather models investigating the impact of wind farms have a horizontal grid too coarse to resolve wind turbines explicitly." P2L10

Page 2 – Line 17: . . . both WFPs delivers. . . -> deliver (no s)

Good catch! P2L16

Page 2 – Line 28: . . . we want to know -> the reviewer suggests to rewrite: ". . . we try to find answers to the following research questions."

Done. P2L26

Page 3 – Line 8: Unique. . .. -> Do you mean "only"?

No, unique in terms of the unique data set. P3L6

Page 3 – Line 9-10: As we have. . . -> This sentence is difficult to understand. Maybe rewrite: We have a SAR satellite image for one of the cases investigated in this study, a brief. . . .

Done. P3L7-8

Page 3 – Line 13: It would be good to have an introductional sentence here, something like: In the framework of the WIPAFF project, the research aircraft Dornier 128-6 operated by TU Braunschweig was used for measuring the wakes of several large offshore wind farms within several campaigns. In these campaigns, the aircraft flew with a true air speed of.....

We agree and adopted your suggestions. P3L11

Page 3 – Line 15: . . . have a resolution of 0.66m -> do you mean .. provide values ever 0,66m of a flight leg?....

Yes, the airborne measurements provide data values every 0.66 m along the flight leg. For a better understanding we changed the sentence to:

Consequently, our measurements provide a data value every 0.66 m along the flight leg (Platis et al., 2018).

Page 3 – Line 24: . . . have the same pattern. -> Suggestion: . . . were conducted using the same pattern.

Done. P3L25

Page 4 – Figuer 1 (Caption): . . . are in use. -> in operation

Done. P4 Fig. 1

Page 4 – Figure 1 (Caption): . . . in 2017 -> I suggest to rewrite: ". . . during the time period investigated in this study.

Done. P4 Fig.1

Page 5 – Line 2: . . . such as Synthetic Aperture Radar (SAR) . . . -> I suggest to add: 'sattelite-borne Synthetic Aperture Radar'

Done. P5L1

Page 5 – Line 8: . . . repeat cycle of 6 days, -> Do you mean return frequency/period?

Repeat cycle is the terminology used in the satellite community, see for example: https://sentinel.esa.int/web/sentinel/missions/sentinel-1/satellite-description/orbit. Therefore, we stick to repeat cycle. P5L8

Page 5 – Line 9: . . . Sentinel 1 passes in the German bight. . . -> This sentence is hard to understand. I suggest something like: Sentinel 1 passes the German Bight at 5am or 5pm

True. We changed the sentence according to your suggestions. P5L10

Page 5 – Line 17: 16 km in the outermost domain followed by two domains with 5 km and 1.67 km,. . . -> Usally a factor of 3 or 5 is used in WRF simulations. Do you mean 5.3 km? (if yes, please correct throughout the paper)

Important catch, that was simply a typo. It should be 15 km, instead of 16 km. P5L17

Page 5 Line 20-22: The authors reference the study by Hahmann which shows that a spin-up time of more than 12 hours should be used and then don't do this in the third case. Why? If there is no good reason for this, the third case should be re-run with a spinup-time of 12 hours or larger.

Good point. The simulations for case study III, deviated from the observations (Fig. 6, 8, 9 in manuscript), hence we tested several spin-up times to improve the simulations. We obtained best results when initializing the model at 00:00 UTC. We added this information in the corresponding paragraph: "Case study III took place earlier, hence, we originally started the simulations on 14 October 12:00~UTC. However, we obtained better results initializing the mode at 00:00~UTC the night before, resulting in a spin up time of less than 12~hours." P5L23-25.

Page 7 – Figure 4: It is quite difficult to distinguish wind farms and islands especially close to the coast (e.g. Borkum Riffgat). The reviewer suggests to change the wind farm shapes to some greyish color.

Done. P7 Fig. 4

Page 8 – Line 1: A summary of our sensitivity tests appears in Table 2. –> The reviewer suggests to rewrite: A summary of the setups of our sensitivity tests appears in Table 2.

Done. P8L9

Page 9 – Table 3: Meerwind Südost consists of 80 turbines (see Major Point 1). Also, the reviewer suggests to double check all other information (hub heights/diameters) with the official information from the wind farm operators or turbine manufacturers!

See answers to major comments above. Thanks again for the important catch.

Page 10 – Line 1: "stably stratified" –> The reviewer suggests to rewrite: "slightly stably stratified" because the vertical change of the potential temperature with height over the rotor area is very small.

We agree. P10L11

Page 10 – Line 5 (Fig 1 – black thich line). It took me really long to find the black thick line. Maybe add a marker to the line? Or Markers at start and end? (also for the other lines)

We added markers at start and end points of the vertical climb flights for better visibility. P4 Fig. 1

Page 10 –Line 6-7: approx. 2 K warmer than the air temperature –> This is really a lot and thus should be explained. The sensors at FINO1 might be impacted by algae growths etc. You should check with other BSH buoys in the German Bight and a high- resolution sea surface dataset e.g. the OSTIA data if this measurement at FINO1 is really valid. This is also true for case III (see below).

See answers to major comments above.

Page 10 – Line 14-15:Further to the north. . . -> This is true but Sandbank is also very "thin" for westerly winds (only 2 rows of turbines). This point should be added to the discussion.

Great point! We added a comment making your point clear in the observation section: "However, Sandbank has only two turbine rows, hence, we expect generally weaker wakes during westerly winds at the wind farm Sandbank." P10L25-26

Page 10 – Line 21-22: However an SST of 288.5 K was measured at FINO1. . .. (see second last comment above). If it holds to be correct, a discussion should be added that the stability could also be an advective feature coming from another location.

Good point, but there is already a paragraph right in the beginning of the discussion making this point. However, we stress the temperature difference one more time in this paragraph and added the following sentence: "In contrast, the upwind conditions were not captured for the case studies I and III. Both cases were characterized by a large scale flow modified by the land upwind and stable conditions although the SST was warmer than the air temperature." P20L26-28

Page 10 Line 25: Wind speed over. . .. –> The reviewer suggests to correct to "above". This occurs several times in the document.

Done. P14L17

Page 11 – Line 3-4: We suggest that this acceleration emerges due to an enhanced flow around the wind farm due to the stably stratified atmosphere –> This is of course correct but needs further explanation. The reviewer suggests to add something like: This stability hinders the flow to extend vertically above the wind farm and forces it to flow around the wind farm leading to a speed up at the wind farm edges.

Thanks for making this point clearer.

Page 11 – Line 12: TKE over and next to the wind farm -> Suggestion: TKE above and next to the wind farm

Done. P15L3

Page 11 – Line 18: . . . where the shear was greatest (shown in Fig. 8a) –> The reviewer suggests to change to "largest"

Done. P13L9

Page 12 – Line 6: . . . where the horizontal wind shear was greatest –>The reviewer suggests to change to "largest"

Done. P13L18

Page 13 – Line 3: In every case -> The reviewer suggests to change to "In summary, in every case . . .."

Done. P13L22

Page 13 – Line 21: "interpolating the wind speed to a height of 10 m is difficult." –> The WRF model provides the output of U10/V10 which are of course not from a model level but should give comparable results to the 10m wind speed. The authors thus should use these data

It is true that WRF provides a 10-m Wind. However, the WRF 10-m Wind largely disagrees with the SAR observations. Therefore, we use simply the model level closest to 10-m.

Page 13 – Line 28: "of the atmosphere in the model did not change with time." –> better: over time

Done. P14L15

Page 14 – Line 2: "at 200 m AMSL over and next to the wind farms" –> better: above and next to the wind farm

Done. P14L18

Page 14 – Line 4: "A weak increase in wind speed similar. . ." –> What this the physical explanation for this increase?

Due to the shorter distance of the upwind edge of the wind farm as stated in the manuscript: "A weak increase in wind speed similar to the observation is represented over the wind farm (i.e. within the gray shaded area in Fig. 8a), associated to the shorter distance of the upwind edge of the wind farm" P14L20

Page 15 – Figure 8 (also 10,12) –> The reviewer suggests to change the x- axis to kilometres from wind farm center and define a point in the center of the wind farm. This way it is very difficult to estimate what the extension of effects such as the speed up are.

We agree and we tried to follow your suggestions. However, every flight was not exactly along the same path (see Fig. 2), that makes it really hard to convert the lat, lon coordinates to a distance as a fixed reference point is missing. Therefore, we stick to latitude coordinates.

Page 15 – Line 6: "showed an increased impact on the wind speed" –> Do you mean "showed an increase of the wind speed above the wind farm"?

No. We reformulated this sentence for clarity: "However, at 10:00~UTC, the observation showed a more pronounced decreased wind speed over the wind farms similar" P14L33-P15L35

Page 16 – Line 5: "for the simulations conducted for 14 October 2017" –> The reviewer suggests: "for the simulations of 14 October 2017".

Done. P15L10

Page 17 – Line 6: It is very confusing that the CNTRns are not related to the cases here. The reviewer suggests to add this where needed. E.g. change to 'CNTRb – case II'

Good point, that definitely increases the clarity of the manuscript!

Page 17 – Line 10: "active" –> do you mean activated?

Yes, thanks. P17L11

Page 19 – Line 6: "resolution with a disabled TKE source" –> The reviewer suggests to rewrite to "without a TKE source" to make it more general.

Good point. We write:" We conducted three simulations with the TKE source of the Fitch et al. (2012) parameterization switched off…" P19L2

Page 19 – Line 7: "without an explicit turbulence source (Volker et al., 2015)" –> I think it would be really nice and strengthen the work if another case simulated with the model by Volker would be added but I also see that this would probably be a lot of additional work. . ... So maybe add this to the outlook at the end of the paper?

We actually tried very hard to get the parameterization of Volker et al. (2015) running. However, we could not get a stable numerical setup. We tried different compilers and boundary conditions, but we were not successful. Summarized we totally agree, that it would be an important contribution. P25L5

Page 19 – Line 21: ". . .produced too small of TKE compared. . .." –> Do you mean ". . .. produced too small amounts of TKE compared. . ."?

Yes, thanks for the catch. P20L8

Page 19 – Line 24: "in a greatly reduced TKE" -> The reviewer suggests to write "largely reduced TKE".

Done. P20L10

Page 19 – Line 26: "over the wind farm - 2 m s−1" -> Better: "over the wind farm of − 2 ms-1"

Done. P20L12

Page 22 – Line 10: "Strong shear lines at the edge of a wind farm or wake. . ." -> The reviewer suggests to add: that originate in the flow around the wind farm.

Done. P23L9-8

Page 22 –Line 16: "In model the wind farms GW" –> In the model the wind farms GW

I cannot see any difference between the actual version and the suggested improvement…, sorry

Page 23 – Line 24: "but feasible for regional climate simulations." -> The reviewer suggests to add "weather simulations"

Done. P24L24

Page 24 – Line 8: The reviewer suggests to change this last sentence of the Conclusions: Thus, future work on wake effects of large offshore wind farms should primarily focus on

Done. P25L9

Page 28 – Line 23-24: Svensson et al –> doi missing

This article is actually not having a DOI, at least we could not find one.

**Reviewer #3**

**Summary**

The authors of the manuscript studied the atmospheric boundary layer above offshore wind farms for three cases. Co-located aircraft measurements are available and horizontal wind speed and TKE are measured. In particular their sensitivity study of different setups regarding the horizontal and vertical resolution, influence of turbulence source terms, and turbulence advection. Having high resolution reference measure- ments makes this work useful as it can go beyond stating differences between model setups. The manuscript is well written and concise. I recommend publication but have some minor points that could make the manuscript easier to follow for the reader.

**General comments**

- It is hard to follow where the wind is coming from in the plots. The flights are neither cross wind nor along the wind direction but at an angle. This needs to be clarified earlier as it took me until page 14 to understand this. E.g. Fig 8: It is not really clear where the wake is.

  During all observation, the flight legs are actually orientated more or less perpendicular to the dominating wind direction. This is also clearly stated in section 2.1, describing the aircraft measurements. However, to avoid any confusion here, we added now a sentence to the captions of Fig. 8 and Fig. 9 stating that the flight legs are perpendicular to the mean wind speed. We apologize for this misunderstanding.

- Synoptic forcing is mentioned but not clearly defined where it is taken from.

  The arrows labeled as synoptic forcing are showing the wind direction measured by the aircraft. Hence, synoptic forcing is misleading here. Therefore, we relabeled the arrows with wind direction and state in the caption that the wind direction was derived from the aircraft measurements.

- The naming of the simulations does not make it clear which cases have been simulated. I would presume that all cases would be simulated with all the simulation setups in Tab 2, which would correspondent to 99 different simulations. Has this been done?

  We agree that the terminology was confusing. We added an extra column to Table 2 and we added an extra paragraph: „We conducted three control simulations namely CNTRa, CNTRb and CNTRc corresponding to the three case studies. These three simulations are identical in terms of their numerical setup, they only simulate different days. " (P5L20-24)

- Please check references to Figures carefully, I noticed some that seemed to reference the wrong Figure but might have not caught them all.

  We rearranged the order of figures (Fig. 12 is now Fig. 11 and vice versa) and checked all references. Thanks for this comment.

- Please increase the size of Fig. 4 and 5. They greatly helpful understanding the paper but are hard to read.

  Good point. We increased Fig.4 and Fig. 5 in size. We hope that's fine for the technical editor as we had to change the latex width from 12 cm to 18 cm.

- In the Discussion you give recommendations that can be understood as quite general. They are based on one case, where WRF got the background flow correctly. Please stress this fact more as it poses a clear limitation. A reader who scans the Abstract and Discussion would benefit from this.

  We agree that our recommendations are only based on one case study. However, evaluation data for offshore regions is hard to get and we don't know any other study that actually tested a wind farm parameterization to that extent for a real case. Hence, the recommendations should be understood as general recommendations for simulations representing the impact of offshore wind farms using the wind farm parameterization of Fitch et al. (2012) during stable conditions. However, to clarify this point we write now: "Based on the successful representation of case study II, we suggest that mesoscale wind farm simulations should use…" P22L19

**Specific comments**

- P3, L15 - 17: This sentence is confusing and I am not sure what is meant.

  We agree, this sentence was too long. We rewrote this section and write now: "Wind speed observations have a relative error of 1% during the flights above the wind farms and 10% during the climb flights resulting from different sample sizes, i.e. 3000 data points during the flights above offshore wind farms and 300 data points during climb flights (Platis et al., 2018; Siedersleben et al., 2018b). The errors in the wind speed measurements cause an error of ± 3∘ in the wind direction shown in the vertical profiles in section 3 (Siedersleben et al., 2018b). " P3L15-18

- P5 l33 to P6 L1: The thrust coefficient is related to the thrust of the wind turbine not the energy.

  Yes, that's true. However, in the parameterization of Fitch et al. (2012) the thrust coefficient describes the fraction of kinetic energy extracted from the atmosphere. To clarify this, we changed the sentenced:" the thrust coefficient describes the fraction of energy extracted from the mean flow in the parameterization of Fitch et al. (2012) (see equation (2) in Fitch et al. (2012)), the power coefficient is the fraction of energy converted into electrical energy." P6L4

- Fig. 2. Simulations are mentioned (CNTR) but not introduced beforehand. This might confuse the reader.

  Ok, that's true. However, mentioning the control simulations not in the caption is also senseless. Therefore, we leave the caption of Fig. 2 untouched.

- Eq. 1. In the text at P7L4 to 7 you explain what $V_H$ and $V_{ij}$ are. How is $V_H$ different from $V_{ij}$?

  Thanks for the comment. That was actually wrong. We renamed $V_H$ and $V_{ij}$ to $V_{ij}$ and $V_{ijk}$, respectively, and adapted the equation (1) and text accordingly: "The variable $N^{ij}_t$ describes wind turbine density within a grid cell i, j; $V_{ijk}$ is the horizontal wind speed at a grid cell ijk, intersecting with the rotor area whereby $V_{ij}$ is the wind speed at hub height at grid cell ij. (Redfern et al. (2019) showed that during strong shear events rotor-equivalent wind speed…" P8L4-6

- Fig. 3: Similar issue as before. It is unclear what the simulation setups are at this point.

  Yes, that's true. However, we think writing a paper in a way that one figure is used several times is quite normal.

- P9, Tab 2: Add which cases are simulated. What is the difference between CNTRa, CNTRb, and CNTRc?

  We added an extra paragraph in the numerical setup section to make this clear: „We conducted three control simulations namely CNTRa, CNTRb and CNTRc corresponding to the three case studies. These three simulations are identical in terms of their numerical setup, they only simulate different days." (P5L20-24)

  Additionally, we added an extra column to Table 2, describing the actual case study that was simulated.

- P10, L1-7: How was stratification determined here? The measured wind speed profile does not suggest much shear and the potential temperature profiles shows very little variation.

  We agree that the measured potential temperature profile doesn't show a strong gradient. Therefore, we write now:" Case study I was slightly stably stratified with wind from the south west." P10L9

- Some problems with Figure references (6g is wind direction but temperature is referred). Fig. 1 is referenced but probably Figure 6 meant.

  We added additional references to avoid any confusion on P10L9 to Fig. 6a, g. The reference to Fig. 1 was given because the location of the climb flights is shown in there. To clarify this, we write now: "for location of climb flight see Fig. 1". P10L13

- P10ff: Which simulation from Tab. 1 are presented here? What resolution?

  This is the pure observational data we presenting here. Additionally, we show ERA5 reanalysis data in Fig. 5. To clarify this, we changed the title of the section "Synoptics and mesoscale overview" → "Synoptics and mesoscale overview based on ERA5 reanalysis, SAR and aircraft data"

- Fig. 7: It is hard to make out differences in the wind speed with the chosen wind speed scaling. Please split it by case to make it easier for the reader.

  Done, see Fig. 7 on P15.

- Fig. 8: It is not intuitive to the reader to follow if a wake effect is expected outside of the wind farm region. It would help to add another shade to where a wake effect would be expected. This is linked to the first general comment.

  All aircraft observations were taken over offshore wind farms. Therefore, a wake region in a classical point of view does not exist for these measurements. For further details, see answers to the first general comment.

- P18, L1: Reference to Figure 12. L5: "TKE only $0.3m^2s^{-2}$ lower than the observed mean". Is this the mean over the wind farm location? L6: Which area with high shear? Are the referenced Figures correct? L10: Within Figure 13 there is no measurement for easy comparison. It could be helpful to define an average TKE over the wind farm and present this consistently for all simulations and the observations

  L1Missing reference to Fig. 12; Done. P17L10

  L5: "TKE only $0.3m^2s^{-2}$ lower than the observed mean". Is this the mean over the wind farm location?

  This is the mean resulting from averaging all four flight legs taken over the wind farm.

  L6: Which area with high shear? Are the referenced Figures correct?

  Great catch, a reference was missing. P17L15

  L10: Within Figure 13 there is no measurement for easy comparison. It could be helpful to define an average TKE over the wind farm and present this consistently for all simulations and the observations

  We agree with the first suggestion and to be consistent we also changed Fig. 11 (now Fig. 12) accordingly. However, the second one defining a spatially averaged TKE for each simulation and the observations doesn't contain any additional information about the TKE. As shown in the manuscript the spatial distribution of the TKE is one of the largest weaknesses of parameterization of Fitch et al. (2012), hence, by spatially averaging the TKE, we won't obtain any further information.

- P19, L/: Is the EWP from Volker et. al. 2015 used here or is the turbulence production term turned off in the Fitch parametrization? Please clarify.

  Good point, that sentence wasn't clear. We write now:" For comparison to wind farm parameterizations without an explicit turbulence source (e.g. Volker et al. 2015) we conducted three simulations with the TKE source of the Fitch et al. (2012) parameterization switched off using…"P18L11-12

**Answers to interactive comment of David Ham**

Code availability

GitHub is an excellent development resource, but it is not a suitable permanent archive location. Indeed, GitHub themselves tell you to use Zenodo for this purpose, and provide a very easy mechanism to archive your GitHub repository on Zenodo. See https://guides.github.com/activities/citable-code/. Please use the facility there and cite the resulting DOI. You may, of course, refer to the GitHub repository in addition to the DOI if you wish (though the Zenodo repository will also link to it).

Data availability

ERA5 is a big dataset! You do not seem to refer to exactly which data is used and therefore should be downloaded. Please do so. If it's too cumbersome to put it directly in the paper, you could, for example, provide the details in your model configuration repository.

For full details of current GMD code and data requirements, please see: https://www. geoscientific-model-development.net/about/code_and_data_policy.html

We agree and uploaded all necessary data to Zenodo, including the two python scripts and one bash script for downloading and postprocessing the ERA5 data.

(https://doi.org/10.5281/zenodo.3490732)

Literature

Platis, A., Siedersleben, S. K., Bange, J., Lampert, A., Bärfuss, K., Hankers, R., Cañadillas, B., Foreman, R., Schultz-Stellenfleth, J., Bughsin D., Neumann, T. & Emeis, S. 2018: First in situ evidence of wakes in the far field behind offshore wind farms. Scientific Reports, 8(1), 2163.

Siedersleben, S. K., Platis, A., Lundquist, J. K., Lampert, A., Bärfuss, K., Cañadillas, B., Bange J., Neumann, T., Emeis, S. 2018a: Evaluation of a wind farm parametrization for mesoscale atmospheric flow models with aircraft measurements. Meteorologische Zeitschrift, 27, 401-415.

Siedersleben, S. K., Lundquist, J. K, Platis, A., Lampert, A., Bärfuss, K., Cañadillas, B., Bange J., Neumann, T., Emeis, S. 2018b: Mircrometeorological impacts of offshore wind farms as seen in observations and simulations. Environ. Res. Lett., 12, 124012.

---

## Author Response (AR2)

Answers to the editor corresponding to the manuscript: Turbulent kinetic energy over large offshore wind farms observed and simulated by the mesoscale model WRF (3.8.1)

The comments of the editor and the corresponding answers of the authors are written in black and in green, respectively.

Comments to the Author:
Dear Authors

the manuscript is close to acceptance.

1.) Often read, but still wrong: Temperatures are low/high, they are definitely not cold or warm (in the same way, humidity is not dry and a speed is not fast!). Please check the whole manuscript for this, in particular sec. 3.1 and sec. 6.

Thanks for the catch!

2.) Could you expand sec. 5.3. by a few sentences? For me a few questions remain. Would you normally consider the advection of TKE the better modelling choice? For me it sounds like including the process should make the simulations more realistic. Why do you find the opposite when you look at TKE above windfarms?

Sure! We expanded section 5.3 by the following sentences:

The simulated TKE within the wake is in the order of 0.6-0.8 $m^2$ $s^{-2}$ meaning that the simulated TKE in the wake is more than twice as high than in the undisturbed flow. This finding is in contrast to the observations reported in Platis et al. (2017), they measured lower TKE values within the wake than in the ambient flow during stable conditions. Summarized, although it is expected that the advection of TKE is supposed to improve generally mesoscale simulations, we observed here two drawbacks with respect to the wind farm parameterization of Fitch et al. (2016). Firstly, the TKE above the wind farm was too low associated with too high wind speeds above the wind farm. Secondly, according to airborne observations of Platis et al. (2017) the TKE within the wake is lower than in the ambient flow, in contrast, activating the TKE advection option results in an enhanced TKE within the wake. Therefore, we conclude not using the TKE advection option for wake simulations during stable conditions at offshore sites. P20L14-22

Language issues:
p.1, l.15: has gained?

Yes, I guess so. Done. P1L15

p.5, l.9: at around ..

Done. P5L9

p.8, l.6: A_ikj = A_ijk??

Great catch, thanks! We renamed A_ikj to A_ijk to make the syntax consistent with V_ijk

p.23, l.2: the size

Done. P23L8

p.23, l.14: In THE model, the wind farm WAKE ...? Or do you really mean the wind farm extends ...

Correct, we really mean the wind farm! P23L20

Citation Fraunhofer not Frauenhofer!

Thanks for the catch!